# Random Quadratic Forms with Dependence: Applications to Restricted Isometry and Beyond

**Arindam Banerjee**     **Qilong Gu**     **Vidyashankar Sivakumar**     **Zhiwei Steven Wu**
Department of Computer Science & Engineering, University of Minnesota, Twin Cities
Minneapolis, MN 55455, USA

## Abstract

Several important families of computational and statistical results in machine learning and randomized algorithms rely on uniform bounds on quadratic forms of random vectors or matrices. Such results include the Johnson-Lindenstrauss (J-L) Lemma, the Restricted Isometry Property (RIP), randomized sketching algorithms, and approximate linear algebra. The existing results critically depend on statistical independence, e.g., independent entries for random vectors, independent rows for random matrices, etc., which prevent their usage in dependent or adaptive modeling settings. In this paper, we show that such independence is in fact not needed for such results which continue to hold under fairly general dependence structures. In particular, we present uniform bounds on random quadratic forms of stochastic processes which are conditionally independent and sub-Gaussian given another (latent) process. Our setup allows general dependencies of the stochastic process on the history of the latent process and the latent process to be influenced by realizations of the stochastic process. The results are thus applicable to adaptive modeling settings and also allows for sequential design of random vectors and matrices. We also discuss stochastic process based forms of J-L, RIP, and sketching, to illustrate the generality of the results. [1]

## 1 Introduction

Over the past few decades, a set of key developments in machine learning and randomized algorithms have been relying on uniform large deviation bounds on quadratic forms involving random vectors or matrices. The Restricted Isometry Property (RIP) is a well known and widely studied result of this type, which has had a major impact in high-dimensional statistics [35, 5, 45, 46]. The Johnson-Lindenstrauss (J-L) Lemma is another well known result of this type, which has led to major statistical and algorithmic advances in the context of random projections [25, 2, 23]. Similar substantial developments have been made in several other contexts, including sketching algorithms based on random matrices [49, 26], advances in approximate linear algebra [32, 20], among others.

Such existing developments in one way or another rely on uniform bounds on quadratic forms of random vectors or matrices. Let $\mathcal{A}$ be a set of $(m \times n)$ matrices and $\boldsymbol{\xi} \in \mathbb{R}^n$ be a sub-Gaussian random vector [45, 46]. The existing results stem from large deviation bounds of the following random variable [28]:

$$C_{\mathcal{A}}(\boldsymbol{\xi}) = \sup_{A \in \mathcal{A}} \left| \|A\boldsymbol{\xi}\|_2^2 - E\|A\boldsymbol{\xi}\|_2^2 \right| . \tag{1}$$

Results such as RIP and J-L can then be obtained in a straightforward manner from such bounds by converting the matrix $A$ into a vector $\theta = \text{vec}(A)$ and converting $\boldsymbol{\xi}$ into a suitable random matrix $X$ to get bounds on

$$C_{\Theta}(X) = \sup_{\theta \in \Theta} \left| \|X\theta\|_2^2 - E\|X\theta\|_2^2 \right| , \tag{2}$$

where $\Theta = \{\text{vec}(A) | A \in \mathcal{A}\}$. Results on other domains such as sketching [49, 26] and approximate linear algebra [32, 20] can be similarly obtained. Further, note that such bounds are considerably more general than the popular Hanson-Wright inequality [39, 22] for quadratic forms of random vectors, which focus on a fixed matrix $A$ instead of a uniform bound over a set $\mathcal{A}$.

The key assumption in all existing results is that the entries $\xi_j$ of $\boldsymbol{\xi}$ need to be *statistically independent*. Such independence assumption shows up as element-wise independence of the random vector $\boldsymbol{\xi}$ in quadratic forms like $C_{\mathcal{A}}(\boldsymbol{\xi})$ and row-wise or element-wise independence of the random matrix $X$ in quadratic forms like $C_\Theta(X)$. Existing analysis techniques, typically based on advanced tools from empirical processes [45, 30], rely on such independence to get the large deviation bound. In this paper, we consider a generalization of such existing results by allowing for statistical dependence in $\boldsymbol{\xi}$. In particular, we assume $\boldsymbol{\xi} = \{\xi_j\}$ to be a stochastic process where the marginal random variables $\xi_j$ are conditionally independent and sub-Gaussian given some other (latent) process $F = \{F_j\}$. While hidden Markov models (HMMs) [7] are a simple example of such a setup, with $F$ being the latent variable sequence and $\boldsymbol{\xi}$ being the observations, our setup described in detail in Section 2 allows for far more complex dependencies, and allows for many different types of graphical models connecting $\boldsymbol{\xi}$ and $F$. In Section 2 we discuss two key conditions such graphical models need to satisfy and give a set of concrete examples of graphical models which satisfy the conditions illustrating the flexibility of the setup. Our main result is to establish a uniform large deviation bound for $C_{\mathcal{A}}(\boldsymbol{\xi})$ in (1) where $\boldsymbol{\xi}$ is any stochastic process following the setup outlined in Section 2.

There are two broad implications of our results allowing for dependence in random quadratic forms. First, there are several emerging domains where data collection, modeling and estimation take place adaptively, including bandits learning, active learning, and time-series analysis [4, 40, 31]. The dependence in such adaptive settings is hard to handle, and existing analysis for specific cases goes to great lengths to work with or around such dependence [36, 18, 34]. The general tool we provide for such settings has the potential to simplify and generalize results in adaptive data collection, e.g., our results are applicable to the smoothed analysis of contextual linear bandits considered in [27]. Second, since our results allow for sequential construction of random vectors and matrices by considering what has happened so far, algorithmic approaches such as J-L and sketching would arguably be able to take advantage of such extra flexibility possibly leading to adaptive and more computationally efficient algorithms. In Section 4, we illustrate how such basic results on adaptive regression, RIP, and J-L would look like by allowing for dependence in the random vectors or matrices.

The technical analysis for our main result is a significant generalization of prior analysis on tail behavior of chaos processes [3, 28, 43] for random vectors with i.i.d. elements. To construct a uniform bound on $C_{\mathcal{A}}(\boldsymbol{\xi})$ in (1) for a stochastic process $\boldsymbol{\xi}$ with statistically dependent entries, we decompose the analysis into two parts: 1) bounding the off-diagonal terms of $A^T A$, and 2) bounding the diagonal terms of $A^T A$. Our analysis for the off-diagonal terms is based on two key tools: decoupling [38] and generic chaining [43], both with suitable generalizations from i.i.d. counter-parts to stochastic processes $\boldsymbol{\xi}$. For decoupling, we present a new result on decoupling of quadratic forms of sub-Gaussian stochastic processes $\boldsymbol{\xi}$ satisfying the conditions of our setup. Our result generalizes the classical decoupling result for vectors with i.i.d. entries [38, 28]. For generic chaining, we develop new results of interest in our context as well as generalize certain existing results for i.i.d. random vectors to stochastic processes. While generic chaining, as a technique, does not rely on statistical independence [43], an execution of the chaining argument does rely on an atomic large deviation bound such as the Hoeffding bound for independent elements [28]. In our setting, the atomic deviation bound in generic chaining carefully utilizes conditional independence satisfied by the stochastic process $\boldsymbol{\xi}$. Our analysis for the diagonal terms is based on suitable use of symmetrization, de-symmetrization, and contraction inequalities [8, 29]. However, we cannot use the standard form for symmetrization and de-symmetrization which are based on i.i.d. elements. We generalize the classical symmetrization and de-symmetrization results [8] to stochastic processes $\boldsymbol{\xi}$ in our setup, and subsequently utilize these inequalities to bound the diagonal terms. We present a gentle exposition to the analysis in Section 3 and the technical proofs are all in [6, Appendix]. We have tried to make the exposition self-contained beyond certain key definitions and concepts such as Talagrand's $\gamma$-function and admissible sequence in generic chaining [43].

**Notation.** Our results are for stochastic processes $\boldsymbol{\xi} = \{\xi_j\}$ adapted to another stochastic process $F = \{F_i\}$ with both moment and conditional independence assumptions outlined in detail in Section 2. We will consider conditional probabilities $X_j = \xi_j | f_{1:j}$, where $f_{1:j}$ is a realization of $F_{1:j}$,

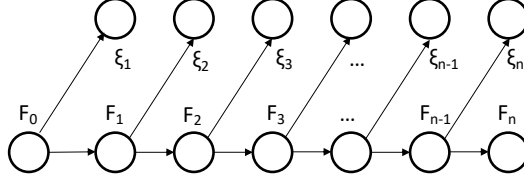

Figure 1: Graphical Model 1 (GM1) structure for stochastic process $\{\xi_i\}$ adapted to $\{F_i\}$ satisfies (SP-2) by construction. While we show arrows only from one random variable, e.g., $F_{i-1} \to \xi_i$, the conditional random variable $\xi_i|F_{1:(i-1)}$ can have dependence on the entire history $F_{1:(i-1)}$. All these arrows are not depicted in this and other figures to avoid clutter.

and assume $X_j$ to be zero-mean $L$-sub-Gaussian, i.e., $\mathbb{P}(|X_j| > \tau) \leq 2\exp(-\tau^2/L^2)$ for some constant $L > 0$ and all $\tau \geq \tau_0$, a constant [45, 46]. For the exposition, we will call a random variable sub-Gaussian without explicitly referring to the constant $L$. With $n$ denoting the length of the stochastic process, we will abuse notation and consider a random vector $\boldsymbol{\xi} = [\xi_j] \in \mathbb{R}^n$ corresponding to the stochastic process $\boldsymbol{\xi} = \{\xi_j\}$, where the usage will be clear from the context. Our results are based on two classes of complexity measures of a set of $(m \times n)$ matrices $\mathcal{A}$. The first class, denoted by $d_F(\mathcal{A})$ and $d_{2\to2}(\mathcal{A})$, are the radius of $\mathcal{A}$ in the Frobenius norm $\|A\|_F = \sqrt{\mathrm{Tr}(A^TA)}$ and the operator norm $\|A\|_{2\to2} = \sup_{\|\mathbf{x}\|_2 \leq 1}\|A\mathbf{x}\|_2$. For the set $\mathcal{A}$, we have $d_F(\mathcal{A}) = \sup_{A\in\mathcal{A}}\|A\|_F$, and $d_{2\to2}(\mathcal{A}) = \sup_{A\in\mathcal{A}}\|A\|_{2\to2}$. The second class is Talagrand's $\gamma_2(\mathcal{A}, \|\cdot\|_{2\to2})$ functional, defined in Section 3 [43, 42]. Recent literature have used the notion of Gaussian width: $w(\mathcal{A}) = E\sup_{A\in\mathcal{A}}|\mathrm{Tr}(G^TA)|$ where $G = [g_{i,j}] \in \mathbb{R}^{m\times n}$ have i.i.d. normal entries, i.e., $g_{i,j} \sim N(0,1)$. It can be shown [43] that $\gamma_2(\mathcal{A}, \|\cdot\|_{2\to2})$ can be bounded by the Gaussian width $w(\mathcal{A})$, i.e., $\gamma_2(\mathcal{A}, \|\cdot\|_{2\to2}) \leq cw(\mathcal{A})$, for some constant $c > 0$. Our analysis will be based on bounding $L_p$-norms of suitable random variables. For a random variable $X$, its $L_p$-norm is $\|X\|_{L_p} = (\mathbb{E}|X|^p)^{1/p}$.

## 2 Setup

We describe the formal set up of stochastic processes for which we provide large deviation bounds. Let $\boldsymbol{\xi} = \{\xi_i\} = \{\xi_1, \ldots, \xi_n\}$ be a sub-Gaussian stochastic process which is decoupled when conditioned on another stochastic process $F = \{F_i\} = \{F_1, \ldots, F_n\}$. In particular, we assume:

(SP-1) for each $i = 1, \ldots, n$, $\xi_i|f_{1:i}$ is a zero mean sub-Gaussian random variable [46] for all realizations $f_{1:i}$ of $F_{1:i}$; and

(SP-2) for each $i = 1, \ldots, n$, there exists an index $\varrho(i) \leq i$ which is non-decreasing, i.e., $\varrho(j) \leq \varrho(i)$ for $j < i$, such that $\xi_i \perp \xi_j|F_{1:\varrho(i)}, j < i$ and $\xi_i \perp F_k|F_{1:\varrho(i)}, k > \varrho(i)$.

where $\perp$ denotes (conditional) independence. The stochastic process $\boldsymbol{\xi} = \{\xi_i\}$ is said to be *adapted to the process $F = \{F_i\}$ satisfying (SP-1) and (SP-2)*.

(SP-1) is an assumption on the moments of the distributions $\xi_i|f_{1:i}$. Note that the assumption allows the specifics of the distribution to depend on the history. (SP-2) is an assumption on the conditional independence structure of $\boldsymbol{\xi}$. The assumption allows $\xi_i$ to depend on the history $F_{1:\varrho(i)}$. Further, we can have $F_{i-1}$ depending on $\xi_{i-1}$ and $\xi_i$ depending on $F_{i-1}$. Graphical models GM1 (Figure 1), GM2 (Figure 2) and GM3 (Figure 3) are examples of graphical models satisfying (SP-2). For GM1, $\varrho(i) = i - 1$ and $F_i$ depends on $F_{1:(i-1)}$, but not on $\xi_i$. Further, $\xi_i$ can depend on the entire history $F_{1:(i-1)}$. GM2 is a variant of GM1 and structurally resembles a HMM (hidden Markov model) with $\varrho(i) = i$, $F_i$ depending on $F_{i-1}$ (or the entire history $F_{1:(i-1)}$), and $\xi_i$ depends on $F_i$ (or the entire history $F_{1:i}$). GM3 is a more complex model with $\varrho(i) = i$ and $F_i$ depends both on $F_{1:(i-1)}$ and $\xi_i$. For GM1 and GM3, we consider an additional 'prior' $F_0$, and the properties (SP-1) and (SP-2) can be naturally extended to include such a prior. We also give concrete examples of potential interest in the context of machine learning in Section 4. For certain graphical models, it may be at times more natural to first construct a stochastic process $\{Z_i\}$ respecting the graphical model structure governed by (SP-2), and then construct the sequence $\{\xi_i\}$ by conditional centering, i.e., $\xi_i|F_{1:i} = Z_i|F_{1:i} - \mathbb{E}[Z_i|F_{1:i}]$ so that $\mathbb{E}[\xi_i|F_{1:i}] = 0$ as required by (SP-1). Such a centered construction is inspired by how one can construct martingale difference sequences (MDS) from martingales [48].

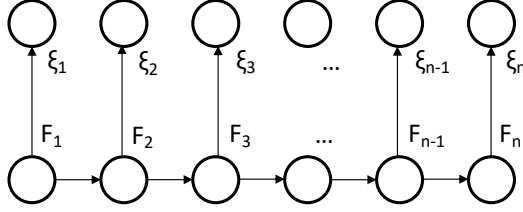

Figure 2: Graphical Model 2 (GM2) structure for stochastic process $\{\xi_i\}$ adapted to $\{F_i\}$ satisfies (SP-2) by construction. While we show arrows only from one random variable, e.g., $F_i \to \xi_i$, the conditional random variable $\xi_i | F_{1:i}$ can have dependence on the entire history $F_{1:i}$.

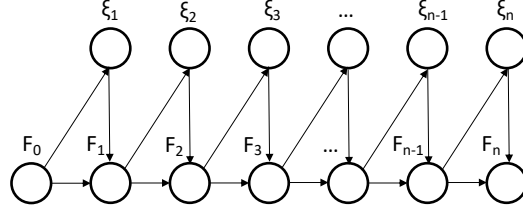

Figure 3: Graphical Model 3 (GM3) structure for stochastic process $\{\xi_i\}$ adapted to $\{F_i\}$ satisfies (SP-2) by construction. Note that there is no restriction on the conditional distribution $F_i \mid (F_{1:(i-1)}, \xi_i)$, so that $F_i$ can have arbitrary dependence on $F_{1:(i-1)}$ and $Z_i$. While we show arrows only to one random variable, e.g., $F_{i-1} \to \xi_i$, the conditional random variable $\xi_i | F_{1:(i-1)}$ can have dependence on the entire history $F_{1:(i-1)}$. Similarly, $F_i | F_{1:(i-1)}, Z_i$ is illustrated only with arrows from $F_{i-1}, Z_i$ to $F_i$ to avoid clutter.

## 3 Main Results

Let $\mathcal{A}$ be a set of $(m \times n)$ matrices and let $\boldsymbol{\xi}$ be a $L$-sub-Gaussian random vector. The random variable of interest for the current analysis is:

$$C_{\mathcal{A}}(\boldsymbol{\xi}) \triangleq \sup_{A \in \mathcal{A}} \left| \|A\boldsymbol{\xi}\|_2^2 - E\|A\boldsymbol{\xi}\|_2^2 \right| . \tag{3}$$

Based on the literature on empirical processes and generic chaining [43, 30], the random variable $C_{\mathcal{A}}(\boldsymbol{\xi})$ can be referred to as an order-2 sub-Gaussian chaos [43, 28]. While widely used results like the restricted isometry property (RIP) [10, 19] and Johnson-Lindenstrauss (J-L) lemma [25, 49] do not explicitly appear in the above form, getting such results from a large deviation bound on $C_{\mathcal{A}}(\boldsymbol{\xi})$ is straightforward [28, 33]. For ease of exposition, we will refer to such converted but otherwise equivalent form as the random matrix form of $C_{\mathcal{A}}(\boldsymbol{\xi})$.

### 3.1 The Main Result: Warm-up

The main technical result in the paper is a large deviation bound on $C_{\mathcal{A}}(\boldsymbol{\xi})$ for the setting when $\boldsymbol{\xi}$ is a stochastic process adapted to $F$ satisfying (SP-1) and (SP-2), as defined in Section 2. To develop large deviation bounds on $C_{\mathcal{A}}(\boldsymbol{\xi})$, we decompose the quadratic form into terms depending on the off-diagonal and the diagonal elements of $A^T A$ respectively. First note that the contributions from the off-diagonal terms of $A^T A$ to $\mathbb{E}\|A\boldsymbol{\xi}\|_2^2$ is 0. To see this, with $A_j$ denoting the $j^{th}$ column of $A$, by linearity of expectation we have

$$\mathbb{E}_{\boldsymbol{\xi}} \left[ \sum_{\substack{j,k=1 \\ j \neq k}}^{n} \xi_j \xi_k \langle A_j, A_k \rangle \right] = \sum_{\substack{j,k=1 \\ j \neq k}}^{n} \mathbb{E}_{\xi_j, \xi_k} [\xi_j \xi_k] \langle A_j, A_k \rangle = \sum_{\substack{j,k=1 \\ j \neq k}}^{n} \mathbb{E}_{F_{1:n}} \left[ \mathbb{E}_{\xi_j, \xi_k | F_{1:n}} [\xi_j \xi_k] \right] \langle A_j, A_k \rangle$$

$$\overset{(a)}{=} \sum_{\substack{j,k=1 \\ j \neq k}}^{n} \mathbb{E}_{F_{1:n}} \left[ \mathbb{E}_{\xi_j | F_{1:n}} [\xi_j] \mathbb{E}_{\xi_k | F_{1:n}} [\xi_k] \right] \langle A_j, A_k \rangle \overset{(b)}{=} 0 ,$$

where (a) follows since $\xi_j \perp \xi_k | F_{1:n}$ by (SP-2) , and (b) follows since $\mathbb{E}_{\xi_j | F_{1:n}} [\xi_j] = \mathbb{E}_{\xi_k | F_{1:n}} [\xi_k] = 0$ by (SP-1).

Now, by definition and Jensen's inequality, we have

$$C_{\mathcal{A}}(\boldsymbol{\xi}) = \sup_{A \in \mathcal{A}} \left| \|A\boldsymbol{\xi}\|_2^2 - \mathbb{E}\|A\boldsymbol{\xi}\|_2^2 \right|$$

$$= \sup_{A \in \mathcal{A}} \left| \sum_{\substack{j,k=1 \\ j \neq k}}^{n} \boldsymbol{\xi}_j \boldsymbol{\xi}_k \langle A_j, A_k \rangle + \sum_{j=1}^{n} (|\xi_j|^2 - \mathbb{E}|\xi_j|^2)\|A_j\|_2^2 \right|$$

$$\leq \sup_{A \in \mathcal{A}} \left| \sum_{\substack{j,k=1 \\ j \neq k}}^{n} \boldsymbol{\xi}_j \boldsymbol{\xi}_k \langle A_j, A_k \rangle \right| + \sup_{A \in \mathcal{A}} \left| \sum_{j=1}^{n} (|\xi_j|^2 - \mathbb{E}|\xi_j|^2)\|A_j\|_2^2 \right|$$

$$= B_{\mathcal{A}}(\boldsymbol{\xi}) + D_{\mathcal{A}}(\boldsymbol{\xi})$$

Therefore, for any $p \in [1, \infty)$, we have

$$\|C_{\mathcal{A}}(\boldsymbol{\xi})\|_{L_p} \leq \|B_{\mathcal{A}}(\boldsymbol{\xi})\|_{L_p} + \|D_{\mathcal{A}}(\boldsymbol{\xi})\|_{L_p} \,. \tag{4}$$

Our approach to getting a large deviation bound for $C_{\mathcal{A}}(\boldsymbol{\xi})$ is based on bounding $\|C_{\mathcal{A}}(\boldsymbol{\xi})\|_{L_p}$, which in turn is based on bounding $\|B_{\mathcal{A}}(\boldsymbol{\xi})\|_{L_p}$ and $\|D_{\mathcal{A}}(\boldsymbol{\xi})\|_{L_p}$. For convenience, we will refer to $B_{\mathcal{A}}(\boldsymbol{\xi})$ as the off-diagonal term and $D_{\mathcal{A}}(\boldsymbol{\xi})$ as the diagonal term. Such bounds lead to a bound on $\|C_{\mathcal{A}}(\boldsymbol{\xi})\|_{L_p}$ of the form

$$\|C_{\mathcal{A}}(\boldsymbol{\xi})\|_{L_p} \leq a + \sqrt{p} \cdot b + p \cdot c, \quad \forall p \geq 1 \,, \tag{5}$$

where $a, b, c$ are constants which do not depend on $p$. Note that by using the moment-generating function and Markov's inequality [48, 44], these $L_p$-norm bounds imply, for all $u > 0$

$$P(|C_{\mathcal{A}}(\boldsymbol{\xi})| \geq a + b \cdot \sqrt{u} + c \cdot u) \leq e^{-u} \,, \tag{6}$$

or, equivalently

$$P(|C_{\mathcal{A}}(\boldsymbol{\xi})| \geq a + u) \leq \exp\left\{ -\min\left( \frac{u^2}{4b^2}, \frac{u}{2c} \right) \right\} \,, \tag{7}$$

which yields the desired large deviation bound.

### 3.2   Upper Bounding $B_{\mathcal{A}}(\boldsymbol{\xi})$ and $D_{\mathcal{A}}(\boldsymbol{\xi})$

The bound on $\|B_{\mathcal{A}}(\boldsymbol{\xi})\|_{L_p}$ is based on two techniques: decoupling [38] and generic chaining [43]. Our main result in decoupling extends the classical result for $\boldsymbol{\xi}$ with i.i.d. entries to stochastic processes $\boldsymbol{\xi}$ satisfying (SP-1) and (SP-2). The second part of the analysis uses generic chaining [43, 42] which is arguably one of the most powerful tools for such analysis. Since we use generic chaining, the results are in terms of Talagrand's $\gamma$-functions [43] defined below.

**Definition 1** For a metric space $(T, d)$, an admissible sequence of $T$ is a collection of subsets of $T$, $\{T_r : r \geq 0\}$, with $|T_0| = 1$ and $|T_r| \leq 2^{2^r}$ for all $r \geq 1$. For $\beta \geq 1$, the $\gamma_\beta$ functional is defined by

$$\gamma_\beta(T, d) = \inf \sup_{t \in T} \sum_{r=0}^{\infty} 2^{r/\beta} d(t, T_r) \,, \tag{8}$$

where the infimum is over all admissible sequences of $T$.

In particular, our results are in terms of $\gamma_2(\mathcal{A}, \|\cdot\|_{2\to2})$, which is related to the Gaussian width of the set by the majorizing measure theorem [42, Theorem 2.1.1][43, Theorem 2.4.1]. Recent years have seen major advances in using Gaussian width for both statistical and computational analysis in the context of high-dimensional statistics and related areas [11, 5, 37, 13]. Hence, recent tools for bounding Gaussian width [11, 13] can be applied to our setting to get concrete bounds for cases of interest. For example, if $\mathcal{A}$ is a set of $s$-sparse $(m \times n)$ matrices, $\gamma_2(\mathcal{A}, \|\cdot\|_{2\to2}) \leq c\sqrt{s\log(mn)}$, for some constant $c$ [30, 45] (also see Section 4).

While the diagonal term $D_{\mathcal{A}}(\boldsymbol{\xi})$ does not have any interaction terms of the form $\xi_j\xi_k$, the term depends on centered random variables $|\xi_j|^2 - \mathbb{E}|\xi_j|^2$. Our analysis relies on three key results: symmetrization, de-symmetrization, and contraction [8, 29]. Our overall approach reduces to showing that upper bounds on $D_{\mathcal{A}}(\boldsymbol{\xi})$ can be derived from upper bounds on $D_{\mathcal{A}}(\mathbf{g})$, where $\mathbf{g}$ has i.i.d. normal entries, and additional terms which can be bounded using generic chaining [43]. Bounds on $D_{\mathcal{A}}(\mathbf{g})$ can be obtained using existing results [28].

### 3.3 The Main Result

Based on the analysis above, we have our main result as stated below

**Theorem 1** *Let $\mathcal{A}$ be a set of $(m \times n)$ matrices and let $\boldsymbol{\xi}$ be a stochastic process adapted to $F$ satisfying (SP-1) and (SP-2). Let*

$$M = \gamma_2(\mathcal{A}, \|\cdot\|_{2\to2}) \cdot \left( \gamma_2(\mathcal{A}, \|\cdot\|_{2\to2}) + d_F(\mathcal{A}) \right) \tag{9}$$

$$V = d_{2\to2}(\mathcal{A}) \cdot \left( \gamma_2(\mathcal{A}, \|\cdot\|_{2\to2}) + d_F(\mathcal{A}) \right) \tag{10}$$

$$U = d_{2\to2}^2(\mathcal{A}) . \tag{11}$$

*Then, for any $\varepsilon > 0$,*

$$P\left( \sup_{A \in \mathcal{A}} \left| \|A\boldsymbol{\xi}\|_2^2 - \mathbb{E}\|A\boldsymbol{\xi}\|_2^2 \right| \geq c_1 M + \varepsilon \right) \leq 2\exp\left( -c_2 \min\left\{ \frac{\varepsilon^2}{V^2}, \frac{\varepsilon}{U} \right\} \right) , \tag{12}$$

*where $c_1, c_2$ are constants which depend on the support.*

It is instructive to compare our bounds for stochastic processes $\boldsymbol{\xi}$ satisfying (SP-1) and (SP-2) to the sharpest existing bound on $C_\mathcal{A}(\boldsymbol{\xi})$ for the special case when $\boldsymbol{\xi}$ has i.i.d. sub-Gaussian entries [28]. For this i.i.d. sub-Gaussian case, [28] showed a large deviation bound based on

$$M' = \gamma_2(\mathcal{A}, \|\cdot\|_{2\to2}) \cdot \left( \gamma_2(\mathcal{A}, \|\cdot\|_{2\to2}) + d_F(\mathcal{A}) \right) + d_F(\mathcal{A}) \cdot d_{2\to2}(\mathcal{A}) \tag{13}$$

$$V' = d_{2\to2}(\mathcal{A}) \cdot \left( \gamma_2(\mathcal{A}, \|\cdot\|_{2\to2}) + d_F(\mathcal{A}) \right) \tag{14}$$

$$U' = d_{2\to2}^2(\mathcal{A}) . \tag{15}$$

By comparing the terms with those in Theorem 1, we note that $U = U'$ and $V = V'$ and while $M'$ has an extra additional term $d_F(\mathcal{A}) \cdot d_{2\to2}(\mathcal{A})$, for symmetric sets $\mathcal{A}$ with $\mathcal{A} = -\mathcal{A}$ we have $d_{2\to2}(\mathcal{A}) \leq \gamma_2(\mathcal{A}, \|\cdot\|_{2\to2})$, so the terms are of the same order. Thus, the generalization to the stochastic process $\boldsymbol{\xi}$ yields the same order bound as the i.i.d. case which allows seamless extension of applications of the result to random vectors/matrices with statistical dependence.

Finally, our results can be extended to the case of non-zero mean stochastic processes. In particular with $\mathbf{x} = \boldsymbol{\xi} + \boldsymbol{\mu}$, where $\boldsymbol{\xi}$ is the stochastic process satisfying (SP-1) and (SP-2) and $\boldsymbol{\mu}$ is the mean vector, i.e., $\mathbb{E}[\mathbf{x}] = \boldsymbol{\mu}$, we have $\|A\mathbf{x}\|^2 - \mathbb{E}\|A\mathbf{x}\|_2^2 = (\|A\boldsymbol{\xi}\|_2^2 - \mathbb{E}\|A\boldsymbol{\xi}\|_2^2) + \langle \boldsymbol{\xi}, 2A^T A\boldsymbol{\mu} \rangle$, where the first term is what we analyze and bound in Theorem 1, and the second term is a linear form of $\boldsymbol{\xi}$. For the uniform bound, the two terms can be separated using Jensen's inequality, the first term can be bounded using Theorem 1 and the second term can be bounded using a standard application of generic chaining using (SP-1) and (SP-2). Thus, mean shifted versions of our results also hold.

## 4 Implications of the Main Results

We show several applications of our results, including the Johnson Lindenstrauss (J-L), Restricted Isometry Property (RIP), and sketching. All proofs can be found in [6, Section 4].

### 4.1 Johnson-Lindenstrauss with Stochastic Processes

Let $X \in \mathbb{R}^{n \times p}$ , $n < p$ and let $\mathcal{A}$ be any set of $N$ vectors in $\mathbb{R}^p$. $X$ is a *Johnson-Lindenstrauss transform* (JLT) [25, 2] if for any $\varepsilon > 0$,

$$(1 - \varepsilon)\|u\|_2^2 \leq \|Xu\|_2^2 \leq (1 + \varepsilon)\|u\|_2^2 \quad \text{for all } u \in \mathcal{A} . \tag{16}$$

JLT is a random projection which embeds high-dimensional data into lower-dimensional space while approximately preserving all pairwise distances [49, 32, 24]. JLT has found numerous applications that include searching for an aproximate nearest neighbor in high-dimensional Euclidean space [23], dimension reduction in data bases [1], learning mixture of Gaussians [15] and sketching [49]. It is

well known that $X = \frac{1}{\sqrt{n}}\tilde{X}$, where $\tilde{X}$ contains standard i.i.d. normal elements, is a JLT with high probability when $n = \Omega(\log N)$ [25]. .

Now denote the element in the $i$-th row and $j$-th column of $\tilde{X}$ as $\tilde{X}_{i,j}$, and the $i$-th row as $\tilde{X}_{i,:}$. Let the entries of $\tilde{X}_{i,j}$ being sequentially generated as follows:

1. Initially, draw the first element $\tilde{X}_{1,1}$ from a zero-mean sub-Gaussian distribution.

2. $\tilde{X}_{i,j}$ is a conditionally 1-sub-Gaussian random variable satisfying $\mathbb{E}[\tilde{X}_{i,j}|f_{i,j}] = 0$. The $f_{i,j}$ are realizations of a stochastic process which can possibly depend on the entries $\{\{\tilde{X}_{i',:}\}_{i'<i}, \{\tilde{X}_{i,j'}\}_{j'<j}\}$.

3. $\tilde{X}_{i,j} \perp \{\{\tilde{X}_{i',:}\}_{i'<i}, \{\tilde{X}_{i,j'}\}_{j'<j}\} \mid f_{i,j}$ and $\tilde{X}_{i,j} \perp \{\{f_{i,j'}\}_{j'>j}, \{f_{i',:}\}_{i'>i}\} \mid f_{i,j}$

The following result is an immediate consequence of Theorem 1

**Corollary 1 (JL)** *Let $X \in \mathbb{R}^{n\times p}$ be a matrix constructed as $X = \frac{1}{\sqrt{n}}\tilde{X}$. If we choose $n = \Omega(\epsilon^{-2}\log N)$, $X$ is a JLT with probability at least $1 - \frac{1}{N^c}$ for a constant $c > 0$.*

## 4.2 Restricted Isometry Property (RIP) with Stochastic Processes

Matrices satisfying Restricted Isometry Property (RIP) are approximately orthonormal on sparse vectors [10, 9]. Let $X \in \mathbb{R}^{n\times p}$ and let $\mathcal{A}$ be the set of all $s$-sparse vectors in $\mathbb{R}^p$. We define matrix $X$ to satisfy RIP with the restricted isometry constant $\delta_s \in (0, 1)$ if for all $u \in \mathcal{A}$,

$$(1 - \delta_s)\|u\|_2^2 \leq \frac{1}{n}\|Xu\|_2^2 \leq (1 + \delta_s)\|u\|_2^2 . \tag{17}$$

Matrices satisfying RIP are of interest in high-dimensional statistics and compressed sensing problems where the goal is to recover a sparse signal $\theta^* \in \mathbb{R}^p$ from limited noisy linear measurements [47, 46]. Sub-Gaussian random matrices with i.i.d. rows, e.g., rows sampled from a $N(0, \sigma^2\mathbb{I}_{p\times p})$ satisfies RIP [10, 9, 35, 5] when $n = \Omega(s\log p)$. But the i.i.d. rows assumption is violated in many practical settings when data is generated adaptively/sequentially. Examples include times-series regression and bandits problems [31, 27], active learning [40, 21] or volume sampling [16, 17]. An application of our new results shows that the i.i.d. assumption is not necessary and design matrices generated from dependent elements also satisfy RIP when $n = \Omega(s\log p)$. For example, the following result holds for matrices $X$ generated similar to matrix $\tilde{X}$ in Section 4.1.

**Corollary 2 (RIP)** *Let $X \in \mathbb{R}^{n\times p}$ be a matrix generated from the process outlined earlier. Then for any $\varepsilon > 0$, if we choose $n = \Omega(\varepsilon^{-2}s\log(2p/s))$, then $\delta_s \leq \varepsilon$ with probability at least $1 - \left(\frac{s}{2p}\right)^{cs}$ for a constant $c > 0$.*

**RIP for adaptively generated rows.** Sequential learning problems like linear contextual bandits involve estimating a parameter with a design matrix whose rows are adversarially generated based on previously observed rows and rewards which are linear functions of the rows. An example is the linear contextual bandit problem considered, e.g., in [27, 41]. The data in any time step $t$ is generated as follows [27, 41]: .

1. Let $\mathcal{H}_{t-1}$ denote historical data observed until time $t-1$. In time step $t-1$ an adaptive adversary $\mathcal{A}_{t-1}$ maps the histories to $k$ contexts $\mu_t^1, \ldots, \mu_t^k$ in $\mathbb{R}^p$ with $\|\mu_t^1\|_2 \leq 1$, i.e., $\mathcal{A}_{t-1} : \mathcal{H}_{t-1} \rightarrow (B_2^p)^k$ where $B_2^p$ represents the unit ball in $p$ dimensions. Nature perturbs the contexts with random Gaussian noise, i.e., $x_t^i = \mu_t^i + g_t^i$ with $g_t^i \sim N(0, \sigma^2\mathbb{I}_{p\times p})$. Now, in the context of GM3, $\mathcal{H}_{t-1} \cup \{x_t^1, \ldots, x_t^k\}$ represents $F_{1:t-1}$.

2. In time step $t$, a learner chooses one among $k$ contexts $\{x_t^1, \ldots, x_t^k\}$ based on historical data $\mathcal{H}_{t-1}$. Let $x_t^{i_t}$ denote the selected context and $g_t^{i_t}$ denote the corresponding Gaussian perturbation. In the context of GM3, we denote the centered Gaussian perturbation $g_t^{i_t} - \mathbb{E}[g_t^{i_t}]$ by $\xi_t$. The learner receives the noisy reward $y_t = \langle x_t^{i_t}, \theta^* \rangle + \omega_t$ where $\omega_t$ is an unknown sub-Gaussian noise. History at time step $t$ is now augmented with the new data, i.e., $\mathcal{H}_t = \mathcal{H}_{t-1} \cup \{\{x_t^1, \ldots, x_t^k\}, x_t^{i_t}, y_t\}$.

The data generation process mirrors GM3 with $F_t$ being a sub-Gaussian process which is influenced by $F_{t-1}$ and $\xi_t$ but generated adaptively by an adversary. $\xi_t$ is a sub-Gaussian random vector chosen by the learner using historical data $\mathcal{H}_{t-1}$ satisfying (SP-2). The algorithm proposed in [41, 27] involves a parameter estimation step in each time step $t$ using the observed contexts and the corresponding rewards $\{x_{t'}^{i_{t'}}, y_{t'}\}, 1 \leq t' \leq t$. With $X_t$ the matrix which has the centered Gaussian perturbations $\xi_1, \ldots, \xi_{t-1}$ as rows, [41, 27] show that $\inf_{u \in \mathbb{R}^p} \mathbb{E}[\|X_t u\|_2^2] \geq t\kappa \|u\|_2^2$ for some constant $\kappa$ depending on the problem parameters and require the following lower bound on the non-asymptotic RIP condition for efficient parameter estimation:

$$\left( \inf_{u \in \mathbb{R}^p} \mathbb{E}[\|X_t u\|_2^2] - \epsilon \right) \|u\|_2^2 \leq \inf_{u \in \mathbb{R}^p} \|X_t u\|_2^2 . \tag{18}$$

Since the data generation follows graphical model GM3, the following Corollary 3 is a direct consequence of Theorem 1

**Corollary 3** *Let $X_t$ be a design matrix generated from the process described above. Then for any $\epsilon > 0$, if we choose $t = \Omega(\epsilon^{-2}\kappa^{-2}p)$, then with probability atleast $1 - \exp(-cp)$ for constant $c > 0$, the following condition is satisfied,*

$$\inf_{u \in \mathbb{R}^p} \|X^t u\|_2^2 \geq t\kappa(1 - \epsilon)\|u\|_2^2 . \tag{19}$$

### 4.3 CountSketch

CountSketch or sparse JL transform is used in real world applications like data streaming and dimensionality reduction [12, 49]. Every column of a $(n \times p)$ CountSketch matrix $X$ has only $d(d \ll n)$ non-zero elements, therefore for any vector $u \in \mathbb{R}^p$, computing $Xu$ takes only $O(dp)$ instead of $O(np)$. Each entry of a CountSketch matrix $X$ is given by $X_{i,j} = \eta_{i,j}\delta_{i,j}/\sqrt{d}$, where $\delta_{i,j}$ is an independent Rademacher random variable, and $\eta_{i,j}$ is a random variable sampled adaptively. The $\eta_{i,j}$ satisfy $\sum_{i=1}^n \eta_{i,j} = d$, $\eta_{i,j} \in \{0, 1\}$, that is each column has exactly $d$ non zero elements. For every column $j$ of $X$, the $\eta_{i,j}$ can be generated by sampling $d$ indices from $\{1, 2, \ldots, n\}$ adaptively given previous columns, then set corresponding $X_{i,j}$ to be a Rademacher random variable, so that $X_{i,j}$ depends on $X_{1,j}, X_{2,j} \ldots, X_{i-1,j}$. The data generation process of countSketch matrix follows graphical model GM1. The variance of $X_{i,j}$ is $\frac{1}{n}$ and since all the entries of $X$ are bounded by 1, $X$ is a JLT over $N$ points when the number of rows satisfies $n = \Omega(\epsilon^{-2} \log N)$. Unlike [14, 26], our bound does not depend on the choice of $d$. Our bound also matches the state of the art [26].

## 5 Conclusions

Several existing results in machine learning and randomized algorithms, e.g., RIP, J-L, sketching, etc., rely on uniform large deviation bounds of random quadratic forms based on random vectors or matrices. Such results are uniform over suitable sets of matrices or vectors, and have found wide ranging applications over the past few decades. Growing interest in adaptive data collection, modeling, and estimation in modern machine learning is revealing a key limitation of such results: the need for statistical independence, e.g., elementwise independence of random vectors, row-wise independence of random matrices, etc. In this paper, we have presented a generalization of such results that allows for statistical dependence on the history. We have also given examples for certain cases of interest, including RIP, J-L, and sketching, illustrating that in spite of allowing for dependence, our bounds are of the same order as that for the case of independent random vectors. We anticipate our results to simplify and help make advances in analyzing learning settings based on adaptive data collection. Further, the added flexibility of designing random matrices sequentially may lead to computationally and/or statistically efficient random projection based algorithms. In future work, we plan to investigate applications of these results in adaptive data collection and modeling settings.

**Acknowledgements:** The research was supported by NSF grants OAC-1934634, IIS-1908104, IIS-1563950, IIS-1447566, IIS-1447574, IIS-1422557, CCF-1451986, a Google Faculty Research Award, a J.P. Morgan Faculty Award, and a Mozilla research grant. Part of this work completed while ZSW was visiting the Simons Institute for the Theory of Computing at UC Berkeley.

## Footnotes

[1]The full version of this paper is available at `https://arxiv.org/abs/1910.04930` [6].

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
