[Supplementary Material]

## A  Decoupling for Martingales

The following definitions are from [36, Chapter 6].

**Definition 2** Let $\{e_i\}$ and $\{d_i\}$ be two sequences of random variables adapted to the $\sigma$-fields $\{\mathcal{F}_i\}$. Then $\{e_i\}$ and $\{d_i\}$ are tangent with respect to $\{\mathcal{F}_i\}$ if, for all $i$,

$$p(d_i|\mathcal{F}_{i-1}) = p(e_i|\mathcal{F}_{i-1}) , \tag{29}$$

where $p(d_i|\mathcal{F}_{i-1})$ denotes the conditional probability of $d_i$ given $\mathcal{F}_{i-1}$.

**Definition 3** A sequence $\{e_i\}$ of random variables adapted to an increasing sequence of $\sigma$-fields $\mathcal{F}_i$ contained in $\mathcal{F}$ is said to satisfy the CI condition (conditional independence) if there exists a $\sigma$-algebra $\mathcal{G}$, contained in $\mathcal{F}$ such that $\{e_i\}$ is conditionally independent given $\mathcal{G}$, and $p(e_i|\mathcal{F}_{i-1}) = p(e_i|\mathcal{G})$.

**Definition 4** A sequence $\{e_i\}$ which satisfies the CI condition and which is also tangent to $\{d_i\}$ is said to be a decoupled tangent sequence to $\{d_i\}$.

The following result is from [36, Proposition 6.1.5].

**Proposition 1** *For any sequence of random variables $\{d_i\}$ adapted to an increasing sequence $\mathcal{F}_i$ of a $\sigma$-algebras, there always exists a decoupled sequence $\{e_i\}$ (on a possibly enlarged probability space) which is tangent to the original sequence and in addition conditionally independent given a master $\sigma$-field $\mathcal{G}$. Frequently $\mathcal{G} = \sigma(\{d_i\})$.*

Next we state our main decoupling result:

**Theorem 3** *Let $\Xi = \{\xi_i\}$ be a martingale difference sequence adapted to an increasing sequence of $\sigma$-fields $\{\mathcal{F}_i\}$. Let $\Xi' = \{\xi'_i\}$ be any decoupled tangent sequence to $\Xi = \{\xi_i\}$. Let $\mathcal{B}$ be a collection of $(n \times n)$ symmetric matrices. Let $F$ be a convex function. Then,*

$$E_{\Xi}\left[\sup_{B \in \mathcal{B}} F\left(\sum_{\substack{j,k=1 \\ j \neq k}}^{n} \xi_j \xi_k B_{j,k}\right)\right] \leq 4 E_{\Xi,\Xi'}\left[\sup_{B \in \mathcal{B}} F\left(\sum_{j,k=1}^{n} \xi_j \xi'_k B_{j,k}\right)\right] . \tag{30}$$

Our proof relies on the following results characterizing distributional equivalence of quadratic forms of tangent sequences. Note that our main result needs decoupled tangent sequences where the additional decoupling property will be used to handle the diagonal terms. We start with the following result:

**Lemma 2** *Let $\Xi = \{\xi_i\}$ be a martingale difference sequence adapted to an increasing sequence of $\sigma$-fields $\{\mathcal{F}_i\}$. Let $\Xi' = \{\xi'_i\}$ be any tangent sequence to $\Xi = \{\xi_i\}$. Let $B$ be a symmetric $(n \times n)$ matrix. Consider the random variables*

$$X_n = \sum_{\substack{j,k=1 \\ j<k}}^{n} \xi_j \xi_k B_{j,k} , \qquad and \qquad X'_n = \sum_{\substack{j,k=1 \\ j<k}}^{n} \xi_j \xi'_k B_{j,k} . \tag{31}$$

*Then $X_n$ and $X'_n$ are identically distributed.*

*Proof:*  We do the proof by induction. When $n = 2$, we have

$$X_2 = \xi_1 \xi_2 B_{1,2} \qquad and \qquad X'_2 = \xi_1 \xi'_2 B_{1,2} .$$

So, the distribution of $X_2$ is

$$
\begin{aligned}
P(X_2 \le x) &= P_{\xi_1,\xi_2}\left(\xi_1\xi_2 B_{1,2} \le x\right) \\
&= P_{\xi_1,\xi_2}\left(\xi_1\xi_2 B_{1,2} \le x, \xi_1 \ge 0\right) + P_{\xi_1,\xi_2}\left(\xi_1\xi_2 B_{1,2} \le x, \xi_1 \le 0\right) \\
&= P_{\xi_1,\xi_2}\left(\xi_2 \le x/(\xi_1 B_{1,2}), \xi_1 \ge 0\right) + P_{\xi_1,\xi_2}\left(\xi_2 \le x/(\xi_1 B_{1,2}), \xi_1 \le 0\right) \\
&= \int_0^\infty p_{\xi_1}(z_1)\left[\int_{-\infty}^{x/(z_1 B_{1,2})} p_{\xi_2|\mathcal{F}_1}(z_2)dz_2\right]dz_1 + \int_{-\infty}^0 p_{\xi_1}(z_1)\left[\int_{x/(z_1 B_{1,2})}^{\infty} p_{\xi_2|\mathcal{F}_1}(z_2)dz_2\right]dz_1 \\
&\overset{(a)}{=} \int_0^\infty p_{\xi_1}(z_1)\left[\int_{-\infty}^{x/(z_1 B_{1,2})} p_{\xi_2'|\mathcal{F}_1}(z_2)dz_2\right]dz_1 + \int_{-\infty}^0 p_{\xi_1}(z_1)\left[\int_{x/(z_1 B_{1,2})}^{\infty} p_{\xi_2'|\mathcal{F}_1}(z_2)dz_2\right]dz_1 \\
&= P_{\xi_1,\xi_2'}\left(\xi_2' \le x/(\xi_1 B_{1,2}), \xi_1 \ge 0\right) + P_{\xi_1,\xi_2'}\left(\xi_2' \le x/(\xi_1 B_{1,2}), \xi_1 \le 0\right) \\
&= P_{\xi_1,\xi_2'}\left(\xi_1\xi_2' B_{1,2} \le x\right) \\
&= P(X_2' \le x)\,,
\end{aligned}
$$

where (a) follows since $\xi_2|\xi_1$ and $\xi_2'|\xi_1$ are identically distributed. Note that for (a), the conditioning is on $\mathcal{F}_1(z_1)$, but we do not show this explicitly. Thus, the statement holds for $n = 2$.

We continue with the proof by induction. Assume that the statement is true for some $m$ so that

$$
X_m = \sum_{\substack{j,k=1 \\ j<k}}^{m} \xi_j \xi_k B_{j,k}\,, \qquad \text{and} \qquad X_m' = \sum_{\substack{j,k=1 \\ j<k}}^{m} \xi_j \xi_k' B_{j,k}
$$

are identically distributed so that

$$
P(X_m \le x) = P(X_m' \le x)\,.
$$

Now, by definition

$$
X_{m+1} = X_m + \xi_{m+1}\sum_{j=1}^{m} \xi_j B_{j,m+1} \qquad \text{and} \qquad X_{m+1}' = X_m' + \xi_{m+1}'\sum_{j=1}^{m} \xi_j B_{j,m+1}\,.
$$

The distribution of $X_{m+1}$ is

$$
\begin{aligned}
P(X_{m+1} \le x) &= P\left(X_m + \xi_{m+1}\sum_{j=1}^{m}\xi_j B_{j,m+1} \le x\right) \\
&= \int_{-\infty}^{\infty} P\left(X_m + \xi_{m+1}\sum_{j=1}^{m}\xi_j B_{j,m+1} \le x \Big| X_m = x_m\right) p_{X_m}(x_m)dx_m \\
&= \int_{-\infty}^{\infty} P\left(\xi_{m+1}\sum_{j=1}^{m}\xi_j B_{j,m+1} \le x - x_m\right) p_{X_m}(x_m)dx_m
\end{aligned}
$$

443 First, note that $p_{X_m}(x_m) = p_{X'_m}(x_m)$ since $X_m$ and $X'_m$ are identically distribution. For the first
444 term, making the random variables explicit, note that

$$P_{\xi_1,\ldots,\xi_m,\xi_{m+1}} \left( \xi_{m+1} \sum_{j=1}^m \xi_j B_{j,m+1} \leq x - x_m \right)$$

$$= P_{\xi_1,\ldots,\xi_m,\xi_{m+1}} \left( \xi_{m+1} \sum_{j=1}^m \xi_j B_{j,m+1} \leq x - x_m, \sum_{j=1}^m \xi_j B_{j,m+1} \geq 0 \right)$$

$$+ P_{\xi_1,\ldots,\xi_m,\xi_{m+1}} \left( \xi_{m+1} \sum_{j=1}^m \xi_j B_{j,m+1} \leq x - x_m, \sum_{j=1}^m \xi_j B_{j,m+1} \leq 0 \right)$$

$$= P_{\xi_1,\ldots,\xi_m,\xi_{m+1}} \left( \xi_{m+1} \leq \frac{x - x_m}{\sum_{j=1}^m \xi_j B_{j,m+1}}, \sum_{j=1}^m \xi_j B_{j,m+1} \geq 0 \right)$$

$$+ P_{\xi_1,\ldots,\xi_m,\xi_{m+1}} \left( \xi_{m+1} \leq \frac{x - x_m}{\sum_{j=1}^m \xi_j B_{j,m+1}}, \sum_{j=1}^m \xi_j B_{j,m+1} \leq 0 \right)$$

445 To simplify notation, let $\chi_m = \sum_{j=1}^m \xi_j B_{j,m+1}$. Note that the distribution of $\chi_m$ depends on $\mathcal{F}_m$,
446 and we explicitly show this dependency as needed in the analysis. Then,

$$P_{\xi_1,\ldots,\xi_m,\xi_{m+1}} \left( \xi_{m+1} \sum_{j=1}^m \xi_j B_{j,m+1} \leq x - x_m \right)$$

$$= P_{\mathcal{F}_m,\xi_{m+1}} \left( \xi_{m+1} \leq \frac{x - x_m}{\chi_m}, \chi_m \geq 0 \right) + P_{\mathcal{F}_m,\xi_{m+1}} \left( \xi_{m+1} \leq \frac{x - x_m}{\chi_m}, \chi_m \leq 0 \right)$$

$$= \int_0^\infty p_{\chi_m}(z_m) \left[ \int_{-\infty}^{(x-x_m)/z_m} p_{\xi_{m+1}|\mathcal{F}_m}(z_{m+1}) dz_{m+1} \right] dz_m$$

$$+ \int_{-\infty}^0 p_{\chi_m}(z_m) \left[ \int_{(x-x_m)/z_m}^\infty p_{\xi_{m+1}|\mathcal{F}_m}(z_{m+1}) dz_{m+1} \right] dz_m$$

$$\overset{(a)}{=} \int_0^\infty p_{\chi_m}(z_m) \left[ \int_{-\infty}^{(x-x_m)/z_m} p_{\xi'_{m+1}|\mathcal{F}_m}(z_{m+1}) dz_{m+1} \right] dz_m$$

$$+ \int_{-\infty}^0 p_{\chi_m}(z_m) \left[ \int_{(x-x_m)/z_m}^\infty p_{\xi'_{m+1}|\mathcal{F}_m}(z_{m+1}) dz_{m+1} \right] dz_m$$

$$= P_{\mathcal{F}_m,\xi'_{m+1}} \left( \xi'_{m+1} \leq \frac{x - x_m}{\chi_m}, \chi_m \geq 0 \right) + P_{\mathcal{F}_m,\xi'_{m+1}} \left( \xi'_{m+1} \leq \frac{x - x_m}{\chi_m}, \chi_m \leq 0 \right)$$

$$= P_{\mathcal{F}_m,\xi'_{m+1}} \left( \xi'_{m+1} \sum_{j=1}^m \xi_j B_{j,m+1} \leq x - x_m \right) .$$

447 That completes the proof. ∎

448 Lemma 2 focuses on the lower triangle of the symmetric matrix $B$. The next result extends the
449 distributional equivalence to the full matrix $B$.

450 **Lemma 3** *Let $\Xi = \{\xi_i\}$ be a martingale difference sequence adapted to an increasing sequence of*
451 *$\sigma$-fields $\{\mathcal{F}_i\}$. Let $\Xi' = \{\xi'_i\}$ be any tangent sequence to $\Xi = \{\xi_i\}$. Let $B$ be a symmetric $(n \times n)$*
452 *matrix. Consider the random variables*

$$Z_n = \sum_{\substack{j,k=1 \\ j \neq k}}^n \xi_j \xi_k B_{j,k} , \qquad \text{and} \qquad Z'_n = \sum_{\substack{j,k=1 \\ j \neq k}}^n \xi_j \xi'_k B_{j,k} . \tag{32}$$

453  *Then $Z_n$ and $Z'_n$ are identically distributed.*

454  *Proof:*  Following Lemma 2, with

$$X_n^{(L)} = \sum_{\substack{j,k=1 \\ j<k}}^{n} \xi_j \xi_k B_{j,k} , \qquad \text{and} \qquad X_n'^{(L)} = \sum_{\substack{j,k=1 \\ j<k}}^{n} \xi_j \xi'_k B_{j,k} , \tag{33}$$

455  we have $X_n \sim X'_n$, i.e., identically distributed. Similarly, with

$$Y_n'' = \sum_{\substack{j,k=1 \\ j>k}}^{n} \xi'_j \xi'_k B_{j,k} , \qquad \text{and} \qquad X_n'^{(U)} = \sum_{\substack{j,k=1 \\ j>k}}^{n} \xi_j \xi'_k B_{j,k} , \tag{34}$$

456  an application of Lemma 2 by interchanging $\Xi = \{\xi\}$ and $\Xi = \{\xi'\}$ implies $Y_n'' \sim X_n'^{(U)}$, and
457  we now provide more details to justify this. First, we switch the notation $j, k$ in (34) and use
458  $B_{j,k} = B_{k,j}$ to get

$$Y_n'' = \sum_{\substack{j,k=1 \\ j<k}}^{n} \xi'_j \xi'_k B_{j,k} , \qquad \text{and} \qquad X_n'^{(U)} = \sum_{\substack{j,k=1 \\ j<k}}^{n} \xi'_j \xi_k B_{j,k} . \tag{35}$$

459  Now, interchanging $\{\xi_j\}$ and $\{\xi'_j\}$, we have

$$Y_n'' = \sum_{\substack{j,k=1 \\ j<k}}^{n} \xi_j \xi_k B_{j,k} , \qquad \text{and} \qquad X_n'^{(U)} = \sum_{\substack{j,k=1 \\ j<k}}^{n} \xi_j \xi'_k B_{j,k} . \tag{36}$$

460  Now $Y_n'' \sim X_n'^{(U)}$ follows from Lemma 2.
461  Continuing with the analysis, since $\Xi$ and $\Xi'$ are tangent sequences, by interchanging $\Xi' = \{\xi'_j\}$ and
462  $\Xi = \{\xi_j\}$ are tangent sequences, with

$$Y_n'' = \sum_{\substack{j,k=1 \\ j>k}}^{n} \xi'_j \xi'_k B_{j,k} , \qquad \text{and} \qquad X_n^{(U)} = \sum_{\substack{j,k=1 \\ j>k}}^{n} \xi_j \xi_k B_{j,k} , \tag{37}$$

463  we have $Y_n'' \sim X_n^{(U)}$. Then, from (34) and (37), we have $X_n^{(U)} \sim X_n'^{(U)}$. Combining this with
464  (33), we have

$$X_n^{(L)} + X_n^{(U)} \sim X_n'^{(L)} + X_n'^{(U)} . \tag{38}$$

465  That completes the proof. ∎

466  *Proof of Theorem 3:* Let $\Delta = \{\delta_1, \ldots, \delta_n\}$ be a set of i.i.d. Bernoulli random variables with $P(\delta_i = 
467  0) = P(\delta_i = 1) = 1/2$. Since $B \in \mathcal{B}$ are symmetric, we have

$$\sum_{\substack{j,k=1 \\ j\neq k}}^{n} \xi_j \xi_k B_{j,k} = 4 E_\Delta \left[ \sum_{\substack{j,k=1 \\ j\neq k}}^{n} \delta_i (1 - \delta_j) \xi_j \xi_k B_{j,k} \right] . \tag{39}$$

By Jensen's inequality

$$F\left(\sum_{\substack{j,k=1\\j\neq k}}^{n}\xi_j\xi_k B_{j,k}\right) = F\left(4E_\Delta\left[\sum_{\substack{j,k=1\\j\neq k}}^{n}\delta_i(1-\delta_j)\xi_j\xi_k B_{j,k}\right]\right)$$

$$\leq 4E_\Delta F\left(\sum_{\substack{j,k=1\\j\neq k}}^{n}\delta_i(1-\delta_j)\xi_j\xi_k B_{j,k}\right)$$

$$\Rightarrow\quad \sup_{B\in\mathcal{B}} F\left(\sum_{\substack{j,k=1\\j\neq k}}^{n}\xi_j\xi_k B_{j,k}\right) \leq 4\sup_{B\in\mathcal{B}} E_\Delta F\left(\sum_{\substack{j,k=1\\j\neq k}}^{n}\delta_i(1-\delta_j)\xi_j\xi_k B_{j,k}\right)$$

$$\Rightarrow\quad E_\Xi\left[\sup_{B\in\mathcal{B}} F\left(\sum_{\substack{j,k=1\\j\neq k}}^{n}\xi_j\xi_k B_{j,k}\right)\right] \leq 4E_\Xi\left[\sup_{B\in\mathcal{B}} E_\Delta F\left(\sum_{\substack{j,k=1\\j\neq k}}^{n}\delta_i(1-\delta_j)\xi_j\xi_k B_{j,k}\right)\right].$$

Consider a fixed realization $\Delta_r = \{\delta_{1,r},\ldots,\delta_{n,r}\}$ of $\Delta$, and consider the subset $I = \{i \in [n]|\delta_{i,r} = 1\}$. Lets $I^c$ be the complement set. Then,

$$4\left[\sum_{\substack{j,k=1\\j\neq k}}^{n}\delta_{i,r}(1-\delta_{j,r})\xi_j\xi_k B_{j,k}\right] = 4\left[\sum_{\substack{(j,k)\in I\times I^c\\j\neq k}}\xi_j\xi_k B_{j,k}\right]. \tag{40}$$

Since $\Xi' = \{\xi_i'\}$ is a tangent sequence to $\Xi = \{\xi_i\}$, by Lemma 3, we have

$$E_\Xi\left[\sup_{B\in\mathcal{B}} F\left(\sum_{\substack{j,k=1\\j\neq k}}^{n}\xi_j\xi_k B_{j,k}\right)\right] \leq 4E_\Xi\left[\sup_{B\in\mathcal{B}} E_\Delta F\left(\sum_{\substack{j,k=1\\j\neq k}}^{n}\delta_i(1-\delta_j)\xi_j\xi_k B_{j,k}\right)\right]$$

$$= 4E_\Xi\left[\sup_{B\in\mathcal{B}} E_\Delta F\left(\sum_{\substack{(j,k)\in I\times I^c\\j\neq k}}\xi_j\xi_k B_{j,k}\right)\right] \tag{41}$$

$$\overset{(a)}{=} 4E_{\Xi,\Xi'}\left[\sup_{B\in\mathcal{B}} E_\Delta F\left(\sum_{\substack{(j,k)\in I\times I^c\\j\neq k}}\xi_j\xi_k' B_{j,k}\right)\right]. \tag{42}$$

where (a) follows from the fact that if two random variables are identically distributed, expectations of the same function applied to them will be the same. The matrix $\hat{B}$ of interest for Lemma 3 here is: $\hat{B}_{j,k} = B_{j,k}$ for $(j,k) \in I \times I^c, j \neq k$ and 0 otherwise. Let

$$Y(\Delta) \triangleq 4\sum_{\substack{j,k=1\\j\neq k\\(j,k)\in I\times I^c}}^{n}\xi_j\xi_k' B_{j,k}\ ,\quad Z(\Delta) \triangleq 4\sum_{\substack{j,k=1\\j\neq k\\(j,k)\notin I\times I^c}}^{n}\xi_j\xi_k' B_{j,k}\ ,\quad W \triangleq 4\sum_{j=1}^{n}\xi_j\xi_j' B_{j,j}\ . \tag{43}$$

By construction, for every realization $\Delta_r$, we have

$$Y(\Delta_r) + Z(\Delta_r) + W = 4\left[\sum_{j,k=1}^{n}\xi_j\xi_k' B_{j,k}\right]. \tag{44}$$

Now, by linearly of expectation, we have

$$E_{\Xi,\Xi'}[Z+W] = 4\sum_{\substack{j,k=1\\j\neq k\\(j,k)\notin I\times I^c}}^{n} E_{\xi_j,\xi_k'}[\xi_j\xi_k']B_{j,k} + 4\sum_{j=1}^{n} E_{\xi_j,\xi_j'}[\xi_j\xi_j']B_{j,j} \ . \tag{45}$$

We focus on one term $E_{\xi_j,\xi_k'}[\xi_j\xi_k']$. For $j < k$, we have

$$E_{\xi_j,\xi_k'}[\xi_j\xi_k'] = E_{\xi_{1:(k-1)}}\left[E_{\xi_j,\xi_k'}\left[\xi_j\xi_k'|\xi_{1:(k-1)}\right]\right] = E_{\xi_{1:(k-1)}}\left[\xi_j E_{\xi_k'}\left[\xi_k'|\xi_{1:(k-1)}\right]\right] = 0 \ ,$$

since $\xi_k'|\xi_{1:(k-1)}$ is a martingale difference sequence, which has zero mean. The argument for $j > k$ is similar by interchanging $\Xi$ and $\Xi'$. Recall that $\Xi, \Xi'$ are decoupled tangent sequences, and, following Proposition 1, let $\mathcal{G} = \sigma(\{\xi_j\})$ be the master $\sigma$-field with respect to which $\{\xi_j'\}$ are conditionally independent. Then, we have

$$E_{\xi_j,\xi_j'}[\xi_j\xi_j'] = E_{\mathcal{G}}\left[E_{\xi_j,\xi_j'}\left[\xi_j\xi_j'|\mathcal{G}\right]\right] \overset{(a)}{=} E_{\mathcal{G}}\left[\xi_j E_{\xi_j'}\left[\xi_j'|\mathcal{G}\right]\right] \overset{(b)}{=} E_{\mathcal{G}}\left[\xi_j E_{\xi_j'}\left[\xi_j'|\mathcal{F}_{j-1}\right]\right] \overset{(c)}{=} 0 \ ,$$

where (a) follows since $\xi_j$ if $\mathcal{G}$-measurable, (b) follows since $P(\xi_j'|\mathcal{G}) = p(\xi_j'|\mathcal{F}_{j-1})$ from Definition 3, and (c) follows since $\xi_j'|\mathcal{F}_{j-1}$ is a MDS. As a result, it follows that

$$E_{\Xi,\Xi'}[Z+W] = 0 \ . \tag{46}$$

Now, for any convex function $H$, we have $E_{\Xi,\Xi'}H(Y) = E_{\Xi,\Xi'}H(Y + E_{\Xi,\Xi'}[Z+W]) \leq E_{\Xi,\Xi'}H(Y+Z+W)$. Then, from (42), we have

$$E_{\Xi}\left[\sup_{B\in\mathcal{B}} F\left(\sum_{\substack{j,k=1\\j\neq k}}^{n}\xi_j\xi_k B_{j,k}\right)\right] \leq 4E_{\Xi,\Xi'}\left[\sup_{B\in\mathcal{B}} E_{\Delta}F\left(\sum_{\substack{(j,k)\in I\times I^c\\j\neq k}}\xi_j\xi_k' B_{j,k}\right)\right]$$

$$\leq 4E_{\Xi,\Xi'}\left[\sup_{B\in\mathcal{B}} E_{\Delta}F\left(\sum_{(j,k)=1}^{n}\xi_j\xi_k' B_{j,k}\right)\right]$$

$$= 4E_{\Xi,\Xi'}\left[\sup_{B\in\mathcal{B}} F\left(\sum_{(j,k)=1}^{n}\xi_j\xi_k' B_{j,k}\right)\right] \ .$$

That completes the proof. ∎

## B  Bounds for Sub-Gaussian MDS

### B.1  Overall Analysis

For a MDS $\boldsymbol{\xi} = \{\xi_j\}$, let

$$C_{\mathcal{A}}(\boldsymbol{\xi}) \triangleq \sup_{A\in\mathcal{A}} \left|\|A\boldsymbol{\xi}\|_2^2 - E\|A\boldsymbol{\xi}\|_2^2\right| \tag{47}$$

$$B_{\mathcal{A}}(\boldsymbol{\xi}) \triangleq \sup_{A\in\mathcal{A}} \left|\sum_{\substack{j,k=1\\j\neq k}}^{n} \boldsymbol{\xi}_j\boldsymbol{\xi}_k\langle A_j, A_k\rangle\right| \tag{48}$$

$$D_{\mathcal{A}}(\boldsymbol{\xi}) \triangleq \sup_{A\in\mathcal{A}} \left|\sum_{j=1}^{n}(|\xi_j|^2 - E|\xi_j|^2)\|A_j\|_2^2\right| \tag{49}$$

First, note that the contributions from the off-diagonal terms of $E\|A\boldsymbol{\xi}\|_2^2$ is 0:

**Proposition 2** *For $j \neq k$, $E_{\xi_j,\xi_k}[\xi_j\xi_k] = 0$.*

492  *Proof:*  For $j < k$, we have

$$E_{\xi_j, \xi_k}[\xi_j \xi_k] = E_{\mathcal{F}_{k-1}}\left[E_{\xi_j, \xi_k}\left[\xi_j \xi_k | \mathcal{F}_{k-1}\right]\right] = E_{\mathcal{F}_{k-1}}\left[\xi_j E_{\xi_k}\left[\xi_k | \mathcal{F}_{k-1}\right]\right] = 0 \, ,$$

493  since $\xi_k | \mathcal{F}_{k-1}$ is a martingale difference sequence, which has zero mean. The proof for $j > k$ is
494  similar by switching the roles of $j$ and $k$.  ∎

495  As a result, we have

$$
\begin{aligned}
C_{\mathcal{A}}(\boldsymbol{\xi}) &= \sup_{A \in \mathcal{A}} \left| \|A\boldsymbol{\xi}\|_2^2 - E\|A\boldsymbol{\xi}\|_2^2 \right| \\
&= \sup_{A \in \mathcal{A}} \left| \sum_{\substack{j,k=1 \\ j \neq k}}^{n} \boldsymbol{\xi}_j \boldsymbol{\xi}_k \langle A_j, A_k \rangle + \sum_{j=1}^{n} (|\xi_j|^2 - E|\xi_j|^2) \|A_j\|_2^2 \right| \\
&\leq \sup_{A \in \mathcal{A}} \left| \sum_{\substack{j,k=1 \\ j \neq k}}^{n} \boldsymbol{\xi}_j \boldsymbol{\xi}_k \langle A_j, A_k \rangle \right| + \sup_{A \in \mathcal{A}} \left| \sum_{j=1}^{n} (|\xi_j|^2 - E|\xi_j|^2) \|A_j\|_2^2 \right| \\
&= B_{\mathcal{A}}(\boldsymbol{\xi}) + D_{\mathcal{A}}(\boldsymbol{\xi})
\end{aligned}
$$

496  Hence,

$$\|C_{\mathcal{A}}(\boldsymbol{\xi})\|_p \leq \|B_{\mathcal{A}}(\boldsymbol{\xi})\|_p + \|D_{\mathcal{A}}(\boldsymbol{\xi})\|_p \, . \tag{50}$$

497  We bound $\|B_{\mathcal{A}}(\boldsymbol{\xi})\|_p$ in Section B.2 (Theorem 4) and bound $\|D_{\mathcal{A}}(\boldsymbol{\xi})\|_p$ in Section B.4 (Theorem 6)
498  to get a bound on $\|C_{\mathcal{A}}(\boldsymbol{\xi})\|_p$ of the form

$$\|C_{\mathcal{A}}(\boldsymbol{\xi})\|_p \leq a + \sqrt{p} \cdot b + p \cdot c \, , \quad \forall p \geq 1 \, . \tag{51}$$

499  Note that these bounds imply, for all $u$

$$P(|C_{\mathcal{A}}(\boldsymbol{\xi})| \geq a + b \cdot \sqrt{u} + c \cdot u) \leq e^{-u} \, , \tag{52}$$

500  or, equivalently

$$P(|C_{\mathcal{A}}(\boldsymbol{\xi})| \geq a + u) \leq \exp\left\{ -\min\left( \frac{u^2}{4b^2}, \frac{u}{2c} \right) \right\} \, , \tag{53}$$

501  which yields the main result. In the sequel, to avoid clutter, we mostly avoid all absolute constants
502  and constants which depend on $L$ for $L$-sub-Gaussian random variables, i.e., we set them to 1, so
503  the key dependencies are clear. We are inspired by similar choices in the related literature [42, 19].

## B.2   The Off-diagonal Terms

505  The main result for the off-diagonal term is the following:

506  **Theorem 4** *Let $\boldsymbol{\xi}$ be a sub-Gaussian MDS. Then,*

$$
\begin{aligned}
\|B_{\mathcal{A}}(\boldsymbol{\xi})\|_p &\leq \gamma_2(\mathcal{A}, \|\cdot\|_{2\to 2}) \cdot \left( \gamma_2(\mathcal{A}, \|\cdot\|_{2\to 2}) + d_F(\mathcal{A}) \right) \\
&\quad + \sqrt{p} \cdot d_{2\to 2}(\mathcal{A}) \cdot \left( \gamma_2(\mathcal{A}, \|\cdot\|_{2\to 2}) + d_F(\mathcal{A}) \right) + p \cdot d_{2\to 2}^2(\mathcal{A}) \, .
\end{aligned}
$$

507

508  Note that from Theorem 3, we have

$$\|B_{\mathcal{A}}(\boldsymbol{\xi})\|_{L_p} \leq \left\| \sup_{A \in \mathcal{A}} \left| \sum_{\substack{j,k=1 \\ j \neq k}}^{n} \xi_j \xi_k' \langle A_j, A_k \rangle \right| \right\|_{L_p} = \left\| \sup_{A \in \mathcal{A}} |\langle A\boldsymbol{\xi}, A\boldsymbol{\xi}' \rangle| \right\|_{L_p} \, . \tag{54}$$

509  Hence our analysis will focus on bounding (54), the $L_p$-norm of the decoupled quadratic form. We
510  start with the following result:

**Lemma 4** *Let $\boldsymbol{\xi}$ be a sub-Gaussian MDS, and $\boldsymbol{\xi}'$ be a decoupled tangent sequence to $\boldsymbol{\xi}$. Then, for every $p \geq 1$,*

$$\left\| \sup_{A \in \mathcal{A}} \langle A\boldsymbol{\xi}, A\boldsymbol{\xi}' \rangle \right\|_{L_p} \leq \gamma_2(\mathcal{A}, \| \cdot \|_{2 \to 2}) \cdot \|N_{\mathcal{A}}(\boldsymbol{\xi})\|_{L_p} + \sup_{A \in \mathcal{A}} \| \langle A\boldsymbol{\xi}, A\boldsymbol{\xi}' \rangle \|_{L_p} , \qquad (55)$$

*where $N_{\mathcal{A}}(\boldsymbol{\xi}) = \sup_{A \in \mathcal{A}} \|A\boldsymbol{\xi}'\|_2$.*

*Proof of Lemma 4:* Without loss of generality, assume $\mathcal{A}$ is finite. Consider the random variable of interest:

$$\Gamma = \sup_{A \in \mathcal{A}} |\langle A\xi, A\boldsymbol{\xi}' \rangle| .$$

Let $\{T_r\}_{r=0}^{\infty}$ be an admissible sequence for $\mathcal{A}$ for which the minimum in the definition of $\gamma_2(\mathcal{A}, \| \cdot \|_{2 \to 2})$ is attained. Let

$$\pi_r A = d_{2 \to 2}(A, T_r) = \operatorname*{argmin}_{B \in T_r} \|B - A\|_{2 \to 2} \qquad \text{and} \qquad \Delta_r A = \pi_r A - \pi_{r-1} A .$$

For any given $p \geq 1$, let $\ell$ be the largest integer for which $2^\ell \leq 2p$. Then, by a direct computation based on a telescoping sum and application of triangle inequality, we have

$$|\langle A\boldsymbol{\xi}, A\boldsymbol{\xi}' \rangle - \langle (\pi_\ell A)\boldsymbol{\xi}, (\pi_\ell A)\boldsymbol{\xi}' \rangle| \leq \underbrace{\left| \sum_{r=\ell}^{\infty} \langle (\Delta_{r+1} A)\boldsymbol{\xi}, (\pi_{r+1} A)\boldsymbol{\xi}' \rangle \right|}_{S_1} + \underbrace{\left| \sum_{r=\ell}^{\infty} \langle (\pi_r A)\xi, (\Delta_{r+1} A)\xi' \rangle \right|}_{S_2} .$$

$$(56)$$

We focus on $S_1$ noting that the analysis for $S_2$ is similar. Let

$$X_r(A) = \langle (\Delta_{r+1} A)\boldsymbol{\xi}, (\pi_{r+1} A)\boldsymbol{\xi}' \rangle .$$

Conditioning $X_r(A)$ on $\boldsymbol{\xi}'$, we note

$$X_r(A) = \langle (\Delta_{r+1} A)\boldsymbol{\xi}, (\pi_{r+1} A)\boldsymbol{\xi}' \rangle = \langle \boldsymbol{\xi}, (\Delta_{r+1} A)^T (\pi_{r+1} A)\boldsymbol{\xi}' \rangle$$

a weighted sum of a sub-Gaussian MDS. Then, a direct application of the Azuma-Hoeffding bound [] gives

$$P \left( |X_r(A)| > u \|(\Delta_{r+1} A)^T (\pi_{r+1} A)\boldsymbol{\xi}'\|_2 \,\bigg|\, \boldsymbol{\xi}' \right) \leq 2 \exp(-u^2/2) .$$

Using $u = t2^{r/2}$, we get

$$P \left( |X_r(A)| > t2^{r/2} \|(\Delta_{r+1} A)^T (\pi_{r+1} A)\xi'\|_2 \,\bigg|\, \xi' \right) \leq 2 \exp(-t^2 2^r/2) .$$

Since

$$\left| (\Delta_{r+1} A)^T (\pi_{r+1} A)\boldsymbol{\xi}' \right| \leq \|\Delta_{r+1} A\|_{2 \to 2} \sup_{A \in \mathcal{A}} \|A\boldsymbol{\xi}'\|_2 .$$

we have

$$P \left( |X_r(A)| > t2^{r/2} \|\Delta_{r+1} A\|_{2 \to 2} \sup_{A \in \mathcal{A}} \|A\boldsymbol{\xi}'\|_2 \,\bigg|\, \boldsymbol{\xi}' \right) \leq 2 \exp(-t^2 2^r/2) .$$

Now, since $|\{\pi_r A : A \in \mathcal{A}\}| = |T_r| \leq 2^{2^r}$, by union bound, we get

$$P \left( \sup_{A \in \mathcal{A}} \sum_{r=\ell}^{\infty} |X_r(A)| > t \left( \sup_{A \in \mathcal{A}} \sum_{r=\ell}^{\infty} 2^{r/2} \|\Delta_{r+1} A\|_{2 \to 2} \right) \cdot \sup_{A \in \mathcal{A}} \|A\boldsymbol{\xi}'\|_2 \,\bigg|\, \xi' \right)$$

$$\leq 2 \sum_{r=\ell}^{\infty} |T_r| \cdot |T_{r+1}| \cdot \exp(-t^2 2^r/2)$$

$$\leq 2 \sum_{r=\ell}^{\infty} 2^{2^{r+2}} \cdot \exp(-t^2 2^r/2)$$

$$\leq 2 \exp(-2^\ell t^2) ,$$

for all $t \geq t_0$, a constant. Noting that

$$\sup_{A \in \mathcal{A}} \sum_{r=\ell}^{\infty} 2^{r/2} \|\Delta_{r+1} A\|_{2\to 2} = \gamma_2(\mathcal{A}, \|\cdot\|_{2\to 2})$$

$$\sup_{A \in \mathcal{A}} \|A\boldsymbol{\xi}'\|_2 = N_{\mathcal{A}}(\boldsymbol{\xi}') ,$$

we have

$$P\left( \sup_{A \in \mathcal{A}} \sum_{r=\ell}^{\infty} |X_r(A)| > t\gamma_2(\mathcal{A}, \|\cdot\|_{2\to 2}) N_{\mathcal{A}}(\boldsymbol{\xi}') \,\middle|\, \boldsymbol{\xi}' \right) \leq 2\exp(-pt^2) ,$$

since $p \leq 2^\ell$ by construction. In other words, with $V(\boldsymbol{\xi}') = \gamma_2(\mathcal{A}, \|\cdot\|_{2\to 2}) N_{\mathcal{A}}(\boldsymbol{\xi}')$, for $t \geq t_0$ we have

$$P\left( S_1 \geq tV(\boldsymbol{\xi}') \,\middle|\, \boldsymbol{\xi}' \right) \leq 2\exp(-pt^2) .$$

Note that

$$\|S_1\|_{L_p}^p = E_{\boldsymbol{\xi},\boldsymbol{\xi}'} S_1^p = E_{\boldsymbol{\xi}'} \int_0^\infty pt^{p-1} P(S_1 > t \mid \boldsymbol{\xi}') dt$$

Note that

$$\int_0^\infty pt^{p-1} P(S_1 > t \mid \boldsymbol{\xi}') dt = c^p V(\boldsymbol{\xi}')^p + \int_{cV(\boldsymbol{\xi}')}^\infty pt^{p-1} P(S_1 > t \mid \boldsymbol{\xi}') dt$$

$$\leq c^p V(\boldsymbol{\xi}')^p + V(\boldsymbol{\xi}')^p \int_c^\infty p\tau^{p-1} P(S_1 > \tau V(\boldsymbol{\xi}')|\boldsymbol{\xi}') d\tau$$

$$\leq c_1^p V(\boldsymbol{\xi}')^p ,$$

where $c \geq t_0, c_1$ are suitable constants with depend on $L$. As a result, $\|S_1\|_{L_p} \leq c_1 V(\boldsymbol{\xi}') = c_1 V(\boldsymbol{\xi})$. The bound on $\|S_2\|_{L_p}$ is the same, and can be derived similarly. As a result

$$\|S_1 + S_2\|_{L_p} \leq c_2 \gamma_2(\mathcal{A}, \|\cdot\|_{2\to 2}) \|N_{\mathcal{A}}(\boldsymbol{\xi})\|_{L_p} \tag{57}$$

Further, since $|\{\pi_\ell A : A \in \mathcal{A}\}| \leq 2^{2^\ell} \leq \exp(2p)$, we have

$$E \sup_{A \in \mathcal{A}} |\langle(\pi_\ell A)\boldsymbol{\xi}, (\pi_\ell A)\boldsymbol{\xi}'\rangle|^p \sum_{A \in T_\ell} E|\langle A\boldsymbol{\xi}, A\boldsymbol{\xi}'\rangle|^p \leq 2^{2p} \sup_{A \in \mathcal{A}} E|\langle A\boldsymbol{\xi}, A\boldsymbol{\xi}'\rangle|^p ,$$

so that

$$\left\| \sup_{A \in \mathcal{A}} |\langle(\pi_\ell A)\boldsymbol{\xi}, (\pi_\ell A)\boldsymbol{\xi}'\rangle| \right\|_{L_p} \leq 4\| \sup_{A \in \mathcal{A}} E|\langle A\boldsymbol{\xi}, A\boldsymbol{\xi}'\rangle| \|_{L_p} . \tag{58}$$

Combining (56), (57), and (58) using triangle inequality completes the proof. ∎

For the first term in Lemma 4, we have the following bound:

**Lemma 5** *Let $\boldsymbol{\xi}$ be a MDS. Then*
$$\|N_{\mathcal{A}}(\boldsymbol{\xi})\|_p \leq \gamma_2(\mathcal{A}, \|\cdot\|_{2\to 2}) + d_F(\mathcal{A}) + \sqrt{p} d_{2\to 2}(\mathcal{A}) . \tag{59}$$

*Proof:* Consider the set $S = \{A^T x : x \in B_2^n, A \in \mathcal{A}\}$. Since $\boldsymbol{\xi}$ is a L-sub-Gaussian MDS, we have

$$\|N_{\mathcal{A}}(\boldsymbol{\xi})\|_{L_p} = (E \sup_{A \in \mathcal{A}, x \in B_2^n} |\langle A\boldsymbol{\xi}, x\rangle|^p)^{1/p} = (E \sup_{u \in S} |\langle \boldsymbol{\xi}, \mathbf{u}\rangle|^p)^{1/p}$$

$$\overset{(a)}{\leq} E \sup_{u \in S} |\langle u, \mathbf{g}\rangle| + \sup_{u \in S}(E|\langle \boldsymbol{\xi}, u\rangle|^p)^{1/p}$$

$$= E \sup_{A \in \mathcal{A}, x \in B_2^n} |\langle A\mathbf{g}, x\rangle| + \sqrt{p} \sup_{A \in \mathcal{A}, x \in B_2^n} \|A^T x\|_2$$

$$= E \sup_{A \in \mathcal{A}} N_{\mathcal{A}}(\mathbf{g}) + \sqrt{p} d_{2\to 2}(\mathcal{A})$$

$$\overset{(b)}{\leq} \gamma_2(\mathcal{A}, \|\cdot\|_{2\to 2}) + d_F(\mathcal{A}) + \sqrt{p} d_{2\to 2}(\mathcal{A}) ,$$

544     where (a) follows from Lemma 7 and (b) follows from [19, Lemma 3.7].     ∎

545     For the second term, we have the following bound:

546     **Lemma 6** *Let $\boldsymbol{\xi}$ be a sub-Gaussian MDS, and $\boldsymbol{\xi}'$ be a decoupled tangent sequence. Then, for every*
547     $p \geq 1$,

$$\sup_{A \in \mathcal{A}} \|\langle A\boldsymbol{\xi}, A\boldsymbol{\xi}' \rangle\|_{L_p} \leq \sqrt{p} d_F(\mathcal{A}) d_{2 \to 2}(\mathcal{A}) + p d_{2 \to 2}^2(\mathcal{A}) . \tag{60}$$

548

549     Proof of Lemma 6 needs the following result:

550     **Lemma 7** *Let $\mathbf{x}_1, \ldots, \mathbf{x}_n \in \mathbb{R}^d$ and $T \subset \mathbb{R}^d$. Let $\boldsymbol{\xi} = \{\xi_j\}$ be a $L$-sub-Gaussian MDS and let*
551     $\mathbf{y} = \sum_{j=1}^n \xi_j \mathbf{x}_j$. *Then, for every $p \geq 1$,*

$$\left( E \sup_{t \in T} |\langle t, \mathbf{y} \rangle|^p \right)^{1/p} \leq c_2 \left( E \left[ \sup_{t \in T} |\langle t, \mathbf{g} \rangle| \right] + \sup_{t \in T} (E|\langle t, \mathbf{y} \rangle|^p)^{1/p} \right) \tag{61}$$

552     *where $c_2$ is a constant which depends on $L$ and $\mathbf{g} = \sum_{j=1}^n g_j x_j$ where $g_i \sim N(0,1)$ are indepen-*
553     *dent.*

554     We need the following basic property of sub-Gaussian random variables [45] to prove Lemma 7.

555     **Proposition 3** *If $X$ is a $L$-sub-Gaussian random variable, then*

$$P(|X| > tL) \leq 2\exp(-t^2), \;\; \forall t \geq 0 \qquad \Leftrightarrow \qquad (E|X|^p)^{1/p} \leq c_0 \sqrt{p} L, \;\; \forall p . \tag{62}$$

556

557     *Proof of Lemma 7.* We assume $T$ is finite without loss of generality. Let $\{T_r\}$ be an optimal
558     admissible sequence of $T$. For any $t \in T$, let $\pi_r(t) = \operatorname{argmin}_{t_r \in T_r} \|t - t_r\|_2$. For any given $p$
559     determining the $p$-norm, choose $\ell$ such that $2^{\ell-1} \leq 2p \leq 2^\ell$, so that $2^\ell/p \leq 4$. Then, by triangle
560     inequality, we have

$$\sup_{t \in T} |\langle t, \mathbf{y} \rangle| \leq \sup_{t \in T} |\langle \pi_\ell(t), \mathbf{y} \rangle| + \sup_{t \in T} \sum_{r=\ell}^{\infty} |\langle \pi_{r+1}(t) - \pi_r(t), \mathbf{y} \rangle| . \tag{63}$$

561     For the first term, note that

$$\begin{aligned}
\left( E \sup_{t \in T} |\langle \pi_\ell(t), \mathbf{y} \rangle|^p \right)^{1/p} &\leq \left( E \sum_{t \in T_\ell} |\langle t, \mathbf{y} \rangle|^p \right)^{1/p} \\
&\leq (|T_\ell|)^{1/p} \sup_{t \in T_\ell} (E|\langle t, \mathbf{y} \rangle|^p)^{1/p} \\
&\leq (2^{2^\ell})^{1/p} \sup_{t \in T} (E|\langle t, \mathbf{y} \rangle|^p)^{1/p} \\
&\leq 16 \sup_{t \in T} (E|\langle t, \mathbf{y} \rangle|^p)^{1/p} .
\end{aligned}$$

562     For the second term, since $\{\xi_j\}$ is a $L$-sub-Gaussuan MDS, we have

$$\begin{aligned}
&P\left( \sup_{t \in T} \sum_{r=\ell}^{\infty} |\langle \pi_{r+1}(t) - \pi_r(t), \mathbf{y} \rangle| \geq uL \sum_{r=\ell}^{\infty} 2^{r/2} \|(\langle \pi_{r+1}(t) - \pi_r(t), \mathbf{x}_j \rangle)_{j=1}^n\|_2 \right) \\
&\leq \sum_{r=\ell}^{\infty} \sum_{t \in T_{r+1}} \sum_{t' \in T_r} P\left( \left| \sum_{j=1}^n \xi_j \langle t - t', \mathbf{x}_j \rangle \right| \geq uL 2^{r/2} \|\langle t - t', \mathbf{x}_j \rangle_{j=1}^n\|_2 \right) \\
&\overset{(a)}{\leq} \sum_{r=\ell}^{\infty} 2^{2^{r+1}} \cdot 2^{2^r} \cdot \exp(-2^r u^2/2) \leq 2\exp(-2^\ell u^2/4) \\
&\leq 2\exp(-pu^2/2) ,
\end{aligned}$$

for $u > c$, a constant (see Remark on generic chaining union bound in the sequel), where (a) follows from Azuma-Hoeffding inequality. Then, from Proposition 3, we have

$$\left( E \sup_{t \in T} \sum_{r=\ell}^{\infty} |\langle \pi_{r+1}(t) - \pi_r(t), \mathbf{y} \rangle|^p \right)^{1/p} \leq L \sum_{r=\ell}^{\infty} 2^{r/2} \|(\langle \pi_{r+1}(t) - \pi_r(t), \mathbf{x}_j \rangle)_{j=1}^n\|_2$$
$$\leq L\gamma_2(T', \|\cdot\|_2) ,$$

where $T' = \{(\langle t, \mathbf{x}_j \rangle)_{j=1}^n | t \in T\}$. Then, by the majorizing measures theorem [42, 41], we have

$$\gamma_2(T', \|\cdot\|_2) \leq E \sup_{t' \in T'} |\langle t', \mathbf{g} \rangle| = E \sup_{t \in T} \left| \sum_{j=1}^n \langle t, \mathbf{x}_j \rangle g_j \right| = E \sup_{t \in T} |\langle t, \mathbf{g} \rangle| .$$

That completes the proof. ∎

Before proceeding further, we show the details of how the union bound works out in generic chaining [42]. We use variants of such union bound analysis several times in our proofs, and this is the only place we show the details. Such analysis is considered standard in the context of generic chain, but as a tool generic chaining is not as widely used.

**Remark: Union bound in generic chaining.** After applying union bound in a generic chaining based analysis, we get a (infinite) sum of the following form:

$$\sum_{r=\ell}^{\infty} 2^{2^{r+1}} \cdot 2^{2^r} \cdot \exp(-2^r u^2/2) = \sum_{r=\ell}^{\infty} 2^{3 \cdot 2^r} \cdot \exp(-2 \cdot 2^r u^2/4)$$
$$= \exp(-2^\ell u^2/4) \sum_{r=\ell}^{\infty} \exp^{(3 \log 2) \cdot 2^r} \cdot \exp(-2 \cdot (2^r - 2^\ell) u^2/4) .$$

Focusing on the exponent, note that

$$(3 \log 2) \cdot 2^r - 2 \cdot 2^r u^2/4 + \cdot 2^\ell u^2/4 < -(r - \ell)$$
$$\Rightarrow \quad -(2^{r+1} - 2^\ell) u^2/2 < -(r - \ell) - (3 \log 2) \cdot 2^r$$
$$\Rightarrow \quad (2^{r+1} - 2^\ell) u^2/2 > (r - \ell) + (3 \log 2) \cdot 2^r$$
$$\Rightarrow \quad u^2/2 > \frac{r - \ell}{(2^{r+1} - 2^\ell)} + \frac{(3 \log 2) \cdot 2^r}{2^{r+1} - 2^\ell} .$$

Note that the last term is a decreasing function of $r$, and the maximum is achieved at $r = \ell$ when we have

$$u^2/2 > (3 \log 2) \quad u > \sqrt{6 \log 2} .$$

Thus, the bound holds for $u > u_0$ for a constant $u_0$. ∎

*Proof of Lemma 6:* For $A \in \mathcal{A}$ set $S = \{A^T A \mathbf{x} : \mathbf{x} \in B_2^p\}$. Since $\boldsymbol{\xi}$ is a $L$ sub-Gaussian MDS, the random variable $\langle \boldsymbol{\xi}, A^T A \boldsymbol{\xi} \rangle$ is a weighted sum of a sub-Gaussian MDS when conditioned on $\boldsymbol{\xi}'$. Then, we have

$$\|\langle A\boldsymbol{\xi}, A\boldsymbol{\xi}' \rangle\|_{L_p} = (E_{\boldsymbol{\xi}, \boldsymbol{\xi}'} |\langle A\boldsymbol{\xi}, A\boldsymbol{\xi}' \rangle|^p)^{1/p}$$
$$= (E_{\boldsymbol{\xi}} \{E_{\boldsymbol{\xi}'|\boldsymbol{\xi}} |\langle \boldsymbol{\xi}', A^T A \boldsymbol{\xi} \rangle|^p\})^{1/p}$$
$$\leq (E_{\boldsymbol{\xi}} [L\sqrt{p}^p \|A^T A \boldsymbol{\xi}'\|_2^p])^p$$
$$\leq L\sqrt{p} \left( E_{\boldsymbol{\xi}} \sup_{y \in S} |\langle y, \boldsymbol{\xi} \rangle|^p \right)^{1/p} .$$

Now, from Lemma 7, we have

$$\left( E_{\boldsymbol{\xi}} \sup_{y \in S} |\langle y, \boldsymbol{\xi} \rangle|^p \right)^{1/p} \leq E_{\mathbf{g}} \sup_{\mathbf{y} \in S} |\langle \mathbf{g}, \mathbf{y} \rangle| + \sup_{\mathbf{y} \in S} (E_{\boldsymbol{\xi}} |\langle \boldsymbol{\xi}, \mathbf{y} \rangle|^p)^{1/p} .$$

581 For the first term, we have

$$E_{\mathbf{g}} \sup_{\mathbf{y} \in S} |\langle \mathbf{g}, \mathbf{y} \rangle| = E_{\mathbf{g}} \|A^T A \mathbf{g}\|_2 \le (E \|A^T A \mathbf{g}\|_2^2)^{1/2} = \|A^T A\|_F \le \|A\|_F \|A\|_{2 \to 2} \ .$$

582 For the second term,

$$\sup_{y \in S} (E|\langle \mathbf{y}, \boldsymbol{\xi} \rangle|^p)^{1/p} = \sup_{z \in B_2^p} (E|\langle A^T A z, \boldsymbol{\xi} \rangle|^p)^{1/p} \le L \sup_{z \in B_2^p} \sqrt{p} \|A^T A z\|_2 = L \sqrt{p} \|A\|_{2 \to 2} \ .$$

583 Plugging these bounds on the two terms back and taking supremum over $A \in \mathcal{A}$ completes the
584 proof. ∎

585 *Proof of Theorem 4:* Let $\boldsymbol{\xi}'$ be a decoupled tanget sequence to the MDS $\boldsymbol{\xi}$. Then we have

$$
\begin{aligned}
\|B_{\mathcal{A}\cdot}(\boldsymbol{\xi})\|_{L_p} &= \sup_{A \in \mathcal{A}} \left| \sum_{\substack{j,k=1 \\ j \ne k}}^{n} \xi_j \xi_j \langle A_j, A_k \rangle \right| \\
&\overset{(a)}{\le} \sup_{A \in \mathcal{A}} \left| \sum_{j,k=1}^{n} \xi_j \xi_j' \langle A_j, A_k \rangle \right| \\
&\overset{(b)}{\le} \gamma_2(\mathcal{A}, \|\cdot\|_{2 \to 2}) \cdot \|N_{\mathcal{A}}(\boldsymbol{\xi})\|_{L_p} + \sup_{A \in \mathcal{A}} \|\langle A\boldsymbol{\xi}, A\boldsymbol{\xi}' \rangle\|_{L_p} \\
&\overset{(c)}{\le} \gamma_2(\mathcal{A}, \|\cdot\|_{2 \to 2}) \cdot \left( \gamma_2(\mathcal{A}, \|\cdot\|_{2 \to 2}) + d_F(\mathcal{A}) \right) \\
&\quad + \sqrt{p} \cdot d_{2 \to 2}(\mathcal{A}) \cdot \left( \gamma_2(\mathcal{A}, \|\cdot\|_{2 \to 2}) + d_F(\mathcal{A}) \right) + p \cdot d_{2 \to 2}^2(\mathcal{A}) \ ,
\end{aligned}
$$

586 where (a) follows from Theorem 3, (b) follows from Lemma 4, and (c) follows from Lemma 5 and
587 6. That completes the proof. ∎

## B.3 The Diagonal Terms: Bounded Random Variables

589 For the diagonal terms coresponding to bounded random variables, we have the following main
590 result:

591 **Theorem 5** *Let $\mathcal{A} \in \mathbb{R}^{m \times n}$ be a collection of $(m \times n)$ matrices. $\xi_1, \ldots, \xi_n$ be a bounded MDS,*
592 *and let $\boldsymbol{\xi} \in \mathbb{R}^n$ denote a vector of these random variables. Consider the random variable*

$$D_{\mathcal{A}}(\boldsymbol{\xi}) = \sup_{A \in \mathcal{A}} \left| \sum_{j=1}^{n} (\xi_j^2 - E|\xi_j|^2) \|A^j\|_2^2 \right| \ , \tag{64}$$

593 *where $A^j$ denotes the $j^{th}$ column of A. Then, we have*

$$\|D_{\mathcal{A}}(\xi)\|_{L_p} \le d_F(\mathcal{A}) \cdot \gamma_2(\mathcal{A}, \|\cdot\|_{2 \to 2}) + \sqrt{p} \cdot d_F(\mathcal{A}) \cdot d_{2 \to 2}(\mathcal{A}) \ . \tag{65}$$

594

595 The main observation here is since $\xi_j$ are bounded, so are $\xi_j^2$, implying $\eta = \xi_j^2 - E|\xi_j|^2$ is also
596 sub-Gaussian, and the sequence $\eta_1, \ldots, \eta_n$ is a sub-Gaussian MDS [45]. Based on this observation,
597 the proof of Theorem 5 relies on the following result bounding $L_p$-norms of the supremum of sub-
598 Gaussian MDSs:

599 **Lemma 8** *Let $\boldsymbol{\zeta} = [\zeta_1, \ldots, \zeta_n]$ be a L-sub-Gaussian MDS and let $T \in \mathbb{R}^n$. Then, for every $p \ge 1$,*

$$\left( E \sup_{t \in T} |\langle t, \boldsymbol{\zeta} \rangle|^p \right)^{1/p} \le c_2 \left( E \left[ \sup_{t \in T} |\langle t, \mathbf{g} \rangle| \right] + \sup_{t \in T} (E|\langle t, \boldsymbol{\zeta} \rangle|^p)^{1/p} \right) \tag{66}$$

600 *where $c_2$ is a constant which depends on L, $\mathbf{g} = [g_j]$ where $g_j \sim N(0, 1)$ are independent identically*
601 *distributed normal random variables.*

602 *Proof:* We assume $T$ is finite without loss of generality. Let $\{T_r\}$ be an optimal admissible se-
603 quence of $T$. For any $t \in T$, let $\pi_r(t) = \operatorname{argmin}_{t_r \in T_r} \|t - t_r\|_2$. For any given $p$ determining the
604 $p$-norm, choose $\ell$ such that $2^{\ell-1} \le 2p \le 2^\ell$, so that $2^\ell/p \le 4$. Then, by triangle inequality, we have

$$\sup_{t \in T} |\langle t, \boldsymbol{\zeta} \rangle| \le \sup_{t \in T} |\langle \pi_\ell(t), \boldsymbol{\zeta} \rangle| + \sup_{t \in T} \sum_{r=\ell}^{\infty} |\langle \pi_{r+1}(t) - \pi_r(t), \boldsymbol{\zeta} \rangle| . \tag{67}$$

605 For the first term, note that

$$\left( E \sup_{t \in T} |\langle \pi_\ell(t), \boldsymbol{\zeta} \rangle|^p \right)^{1/p} \le \left( E \sum_{t \in T_\ell} |\langle t, \boldsymbol{\zeta} \rangle|^p \right)^{1/p}$$

$$\le (|T_\ell|)^{1/p} \sup_{t \in T_\ell} (E|\langle t, \boldsymbol{\zeta} \rangle|^p)^{1/p}$$

$$\le (2^{2^\ell})^{1/p} \sup_{t \in T} (E|\langle t, \boldsymbol{\zeta} \rangle|^p)^{1/p}$$

$$\le 16 \sup_{t \in T} (E|\langle t, \boldsymbol{\zeta} \rangle|^p)^{1/p} .$$

606 For the second term, for any $u \ge 0$, we have

$$P\left( \sup_{t \in T} \sum_{r=\ell}^{\infty} |\langle \pi_{r+1}(t) - \pi_r(t), \boldsymbol{\zeta} \rangle| \ge u L 2^{r/2} \|\pi_{r+1}(t) - \pi_r(t)\|_2 \right)$$

$$\le \sum_{t=\ell}^{\infty} \sum_{t \in T_{r+1}} \sum_{t' \in T_r} P\left( \left| \langle t - t', \boldsymbol{\zeta} \rangle \right| \ge u L 2^{r/2} \|t - t'\|_2 \right)$$

$$\overset{(a)}{\le} \sum_{r=\ell}^{\infty} 2^{2^{r+1}} \cdot 2^{2^r} \cdot \exp(-2^r u^2) \le 2 \exp(-2^\ell u^2)$$

$$\le 2 \exp(-p u^2) ,$$

607 for $u \ge u_0$, a constant, and where (a) follows from the Azuma-Hoeffding inequality.

608 Then, from Proposition 3, we have

$$\left( E \sup_{t \in T} \sum_{r=\ell}^{\infty} |\langle \pi_{r+1}(t) - \pi_r(t), \boldsymbol{\zeta} \rangle|^p \right)^{1/p} \le L \sum_{r=\ell}^{\infty} \left( 2^{r/2} \|\pi_{r+1}(t) - \pi_r(t)\|_2 \right) \le L \gamma_2(T, \|\cdot\|_2) .$$

609 Then, by the majorizing measures theorem [][], we have

$$\gamma_2(T, \|\cdot\|_2) \le E \sup_{t \in T} |\langle t, \mathbf{g} \rangle| ,$$

610 where $\mathbf{g} = [g_j], g_j \sim N(0,1)$. That completes the proof. ∎

611 *Proof of Theorem 5.* Consider the random variable $\boldsymbol{\zeta}^{(A)} = \sum_{j=1}^{n} (\xi^2 - E|\xi_j|^2)\|A^j\|_2^2$. Then, for
612 any $A, B \in \mathcal{A}$, by Azuma-Hoeffding inequality, we have

$$P\left( |\boldsymbol{\zeta}^{(A)} - \boldsymbol{\zeta}^{(B)}| \ge \varepsilon \right) \le 2 \exp\left\{ -\frac{\varepsilon^2}{d_2^2(A,B)} \right\} , \tag{68}$$

613 where

$$d_2(A,B) = \left( \sum_{j=1}^{n} (\|A^j\|_2^2 - \|B^j\|_2^2)^2 \right)^{1/2}$$

$$= \left( \sum_{j=1}^{n} (\|A^j\|_2 - \|B^j\|_2)^2 \cdot (\|A^j\|_2 + \|B^j\|_2)^2 \right)^{1/2}$$

$$\overset{(a)}{\le} \left( \sum_{j=1}^{n} \|A^j - B^j\|_2^2 \cdot (\|A^j\|_2 + \|B^j\|_2)^2 \right)^{1/2}$$

$$\le 2 d_F(\mathcal{A}) \|A - B\|_{2 \to 2} ,$$

where (a) follows from triangle inequality. From the majorizing measure theorem [42] we have $E[\sup_{t \in T} |\langle t, \mathbf{g} \rangle|] \leq \gamma_2(T, d_2)$. Then, from Lemma 8 and Proposition 3, we have

$$\|D_{\mathcal{A}}(\boldsymbol{\xi})\|_{L_p} \leq d_F(\mathcal{A}) \cdot \gamma_2(\mathcal{A}, \|\cdot\|_{2 \to 2}) + \sqrt{p} \cdot d_F(\mathcal{A}) \cdot d_{2 \to 2}(\mathcal{A}) .$$

That completes the proof. ∎

## B.4 The Diagonal Terms: Unbounded sub-Gaussian Random Variables

For the diagonal terms corresponding to unbounded sub-Gaussian random variables, we have the following main result:

**Theorem 6** *Let $\mathcal{A} \in \mathbb{R}^{m \times n}$ be a collection of $(m \times n)$ matrices. $\xi_1, \ldots, \xi_n$ be a sub-Gaussian MDS, and let $\xi \in \mathbb{R}^n$ denote a vector of these random variables. Consider the random variable*

$$D_{\mathcal{A}}(\xi) = \sup_{A \in \mathcal{A}} \left| \sum_{j=1}^{n} (\xi_j^2 - E|\xi_j|^2) \|A^j\|_2^2 \right| , \tag{69}$$

*where $A^j$ denotes the $j^{th}$ column of A. Then, we have*

$$\|D_{\mathcal{A}}(\xi)\|_{L_p} \leq \sqrt{\log n} \cdot d_F(\mathcal{A}) \cdot \gamma_2(\mathcal{A}, \|\cdot\|_{2 \to 2}) + \sqrt{p} \cdot d_F(\mathcal{A}) \cdot d_{2 \to 2}(\mathcal{A}) + p \cdot d_F(\mathcal{A}) \cdot d_{2,\infty}(\mathcal{A}) . \tag{70}$$

The proof of Theorem 6 relies on the following result bounding $L_p$-norms of the supremum of sub-exponential MDS processes: (a)

**Lemma 9** *Let $\boldsymbol{\zeta} = [\zeta_1, \ldots, \zeta_n]$ be a L-sub-exponential MDS and let $T \in \mathbb{R}^n, n \geq 2$. Then, for every $p \geq 1$,*

$$\left( E \sup_{t \in T} |\langle t, \boldsymbol{\zeta} \rangle|^p \right)^{1/p} \leq c_2 \left( \sqrt{\log n} \cdot E \left[ \sup_{t \in T} |\langle t, \mathbf{g} \rangle| \right] + \sup_{t \in T} \left( E|\langle t, \boldsymbol{\zeta} \rangle|^p \right)^{1/p} \right) \tag{71}$$

*where $c_2$ is a constant which depends on L, $\mathbf{g} = [g_j]$ where $g_j$ are independent identically distributed normal random variables, and $\eta = [\eta_j]$ where $\eta_j$ are independent identically distributed exponential random variables.*

We need the following basic property of sub-exponential random variables to prove Lemma 9.

**Proposition 4** *If $X$ is a L-sub-exponential random variable, then*

$$P(|X| > tL) \leq 2 \exp(-t) , \ \forall t \geq 0 \qquad \Leftrightarrow \qquad (E|X|^p)^{1/p} \leq c_0 p L , \ \forall p . \tag{72}$$

We also need the following result on mixed tails:

**Proposition 5** *Consider a random variable $X$ such that*

$$P(|X| > \sqrt{t} L_2 + t L_1) \leq 2 \exp(-t) , \ \forall t \geq 0 . \tag{73}$$

*Then*

$$(E|X|^p)^{1/p} \leq c_0 \sqrt{p} L_2 + p L_1 , \ \forall p \geq 1 . \tag{74}$$

*Proof:* Note that for $\sqrt{t} L_1 \leq t L_1$, we have

$$P(|X| > 2\sqrt{t} L_2) \leq P(|X| > \sqrt{t} L_2 + t L_1) \leq 2 \exp(-t)$$
$$\Rightarrow \quad P(|X| > t) \leq 2 \exp(-t^2 / 4 L_2^2) .$$

639    For $\sqrt{t}L_1 \geq tL_1$, we have

$$P(|X| > 2tL_1) \leq P(|X| > \sqrt{t}L_2 + tL_1) \leq 2\exp(-t)$$
$$\Rightarrow \quad P(|X| > t) \leq 2\exp(-t/2L_1) \ .$$

640    Hence, for all $t \geq 0$

$$P(|X| > t) \leq 2\exp(-\min(t^2/4L_2^2, t/2L_1)) \ . \tag{75}$$

641    Now, recall that for any non-negative random variable $E[Z] = \int_0^\infty P(Z \geq u)du$. Using $Z =$
642    $|X|^p, u = t^p$, we have

$$E|X|^p = \int_0^\infty P(|X| > t)pt^{p-1}dt$$
$$\leq \underbrace{2\int_0^\infty \exp(-t^2/4L_2^2)pt^{p-1}dt}_{\mathcal{I}_1} + \underbrace{2\int_0^\infty \exp(-t/2L_1)pt^{p-1}dt}_{\mathcal{I}_2} \ .$$

643    For the first term $\mathcal{I}_1$, consider change of variables $t_2 = t/2L_2$, so $dt = 2L_2 dt_2$ to give

$$\mathcal{I}_1 = 2 \cdot 2^p L_2^p p \int_0^\infty \exp(-t_2^2)t_2^{p-1}dt_2 \leq 2c_2^p L_2^p p(p)^{p/2} \ ,$$

644    for a suitable constant $c_2$, following Proposition 3 [44]. For the second term $\mathcal{I}_2$, consider change of
645    variables $t_1 = t/2L_1$, so $dt = 2L_1 dt_1$ to give

$$\mathcal{I}_2 = 2 \cdot 2^p L_1^p p \int_0^\infty \exp(-t_1)t_1^{p-1}dt_1 \leq 2c_1^p L_1^p p(p)^p \ ,$$

646    for a suitable constant $c_1$, following Proposition 4 [44]. Taking $p$-th roots and using Jensen's in-
647    qeaulity, we have

$$(E|X|^p)^{1/p} \leq (\mathcal{I}_1)^{1/p} + (\mathcal{I}_2)^{1/p} \leq c_0(\sqrt{p}L_2 + pL_1) \ ,$$

648    for a suitable constant $c_0 > 0$. That completes the proof. ∎

649    We also need the following result from [40]:

650    **Theorem 7** *For any $T \subset \mathbb{R}^n$, we have*

$$E\left[\sup_{t\in T}|\langle t, \boldsymbol{\eta}\rangle|\right] \leq \sqrt{\log n} \cdot E\left[\sup_{t\in T}|\langle t, \mathbf{g}\rangle|\right] \ , \tag{76}$$

651    *where $\mathbf{g} = [g_j]$ where $g_j$ are independent identically distributed normal random variables, and*
652    *$\boldsymbol{\eta} = [\eta_j]$ where $\eta_j$ are independent identically distributed exponential random variables.*

653    *Proof of Lemma 9.* We assume $T$ is finite without loss of generality. Let $\{T_r\}$ be an optimal
654    admissible sequence of $T$. For any $t \in T$, let $\pi_r(t) = \arg\min_{t_r \in T_r} \|t - t_r\|_2$. For any given $p$
655    determining the $p$-norm, choose $\ell$ such that $2^{\ell-1} \leq 2p \leq 2^\ell$, so that $2^\ell/p \leq 4$. Then, by triangle
656    inequality, we have

$$\sup_{t\in T}|\langle t, \boldsymbol{\zeta}\rangle| \leq \sup_{t\in T}|\langle \pi_\ell(t), \boldsymbol{\zeta}\rangle| + \sup_{t\in T}\sum_{r=\ell}^\infty |\langle \pi_{r+1}(t) - \pi_r(t), \boldsymbol{\zeta}\rangle| \ . \tag{77}$$

657    For the first term, note that

$$\left(E\sup_{t\in T}|\langle \pi_\ell(t), \boldsymbol{\zeta}\rangle|^p\right)^{1/p} \leq \left(E\sum_{t\in T_\ell}|\langle t, \boldsymbol{\zeta}\rangle|^p\right)^{1/p}$$
$$\leq (|T_\ell|)^{1/p}\sup_{t\in T_\ell}(E|\langle t, \boldsymbol{\zeta}\rangle|^p)^{1/p}$$
$$\leq (2^{2^\ell})^{1/p}\sup_{t\in T}(E|\langle t, \boldsymbol{\zeta}\rangle|^p)^{1/p}$$
$$\leq 16\sup_{t\in T}(E|\langle t, \boldsymbol{\zeta}\rangle|^p)^{1/p} \ .$$

For the second term, for any $u \geq 0$, we have

$$P\left(\sup_{t \in T} \sum_{r=\ell}^{\infty} |\langle \pi_{r+1}(t) - \pi_r(t), \zeta \rangle| \geq \sqrt{u} L 2^{r/2} \|\pi_{r+1}(t) - \pi_r(t)\|_2 \right.$$
$$\left. + uL2^r \|\pi_{r+1}(t) - \pi_r(t)\|_\infty \right)$$

$$\leq \sum_{t=\ell}^{\infty} \sum_{t \in T_{r+1}} \sum_{t' \in T_r} P\left(\left|\langle t - t', \zeta \rangle\right| \geq \sqrt{u} L 2^{r/2} \|t - t'\|_2 \right.$$
$$\left. + uL2^r \|t - t'\|_\infty \right)$$

$$\overset{(a)}{\leq} \sum_{r=\ell}^{\infty} 2^{2^{r+1}} \cdot 2^{2^r} \cdot \exp(-2^r u) \leq 2 \exp(-2^\ell u)$$

$$\leq 2 \exp(-pu) ,$$

for $u \geq u_0$, a constant, and where (a) follows from the Azuma-Bernstein inequality.
Then, from Proposition 5, we have

$$\left(E \sup_{t \in T} \sum_{r=\ell}^{\infty} |\langle \pi_{r+1}(t) - \pi_r(t), \zeta \rangle|^p\right)^{1/p} \leq L \sum_{r=\ell}^{\infty} \left(2^{r/2} \|\pi_{r+1}(t) - \pi_r(t)\|_2 \right.$$
$$\left. + 2^r \|\pi_{r+1}(t) - \pi_r(t)\|_\infty\right)$$
$$\leq L(\gamma_2(T, \|\cdot\|_2) + \gamma_1(T, \|\cdot\|_\infty)) .$$

Then, by the majorizing measures theorem [][], we have

$$\gamma_2(T, \|\cdot\|_2) \leq E \sup_{t \in T} |\langle t, \mathbf{g} \rangle| , \quad \text{and} \quad \gamma_1(T, \|\cdot\|_\infty) \leq E \sup_{t \in T} |\langle t, \boldsymbol{\eta} \rangle| \overset{(a)}{\leq} \sqrt{\log n} \cdot E \sup_{t \in T} |\langle t, \mathbf{g} \rangle|$$

where (a) follows from Theorem 7. Noting that $(1 + \sqrt{\log n}) \leq 3\sqrt{\log n}$ for $n \geq 2$ completes the proof. ∎

*Proof of Theorem 6.* Consider the random variable $\boldsymbol{\zeta}^{(A)} = \sum_{j=1}^n (\xi^2 - E|\xi_j|^2)\|A^j\|_2^2$. Then, for any $A, B \in \mathcal{A}$, by Azuma-Bernstein inequality, we have

$$P\left(|\boldsymbol{\zeta}^{(A)} - \boldsymbol{\zeta}^{(B)}| \geq \varepsilon\right) \leq 2 \exp\left\{-\min\left(\frac{\varepsilon^2}{d_2^2(A,B)}, \frac{\varepsilon}{d_1(A,B)}\right)\right\} , \tag{78}$$

where

$$d_2(A,B) = \left(\sum_{j=1}^n (\|A^j\|_2^2 - \|B^j\|_2^2)^2\right)^{1/2}$$

$$= \left(\sum_{j=1}^n (\|A^j\|_2 - \|B^j\|_2)^2 \cdot (\|A^j\|_2 + \|B^j\|_2)^2\right)^{1/2}$$

$$\overset{(a)}{\leq} \left(\sum_{j=1}^n \|A^j - B^j\|_2^2 \cdot (\|A^j\|_2 + \|B^j\|_2)^2\right)^{1/2}$$

$$\leq 2d_F(\mathcal{A})\|A - B\|_{2 \to 2} ,$$

where (a) follows from triangle inequality, and

$$
\begin{aligned}
d_1(A, B) &= \left\| \|A^j\|_2^2 - \|B^j\|_2^2 \right\|_\infty \\
&= \left\| (\|A^j\|_2 - \|B^j\|_2)(\|A^j\|_2 + \|B^j\|_2) \right\|_\infty \\
&\overset{(a)}{\leq} 2 d_F(\mathcal{A}) \left\| \|A^j - B^j\|_2 \right\|_\infty \\
&= d_F(\mathcal{A}) \|A - B\|_{2,\infty} \,,
\end{aligned}
$$

where (a) follows from triangle inequality. Also, recall from the majorizing measure theorem that $E[\sup_{t\in T} |\langle t, \mathbf{g}\rangle|] \leq \gamma_2(T, d_2)$. Then, from Lemma 9 and Proposition 5, we have

$$
\|D_{\mathcal{A}}(\boldsymbol{\xi})\|_{L_p} \leq \sqrt{\log n} \cdot d_F(\mathcal{A}) \cdot \gamma_2(\mathcal{A}, \|\cdot\|_{2\to2}) + \sqrt{p} \cdot d_F(\mathcal{A}) \cdot d_{2\to2} + p \cdot d_F(\mathcal{A}) \cdot d_{2,\infty}(\mathcal{A}) \,.
$$

That completes the proof. ∎

# C  Proofs of Theorem 1 and Theorem 2

With the existing bounds on the $L_p$ norms of the off-diagonal and diagonal terms from Section B.1, the proofs of the main results, Theorem 1 and Theorem 2, follow from (51)-(53).

*Proof of Theorem 1:* For bounded random variables, the main result follows by combining the off-diagonal $L_p$ norm bound in Theorem 4 and the diagonal $L_p$ norm bound in Theorem 5 and (51)-(53). ∎

*Proof of Theorem 2:* For unbounded sub-Gaussian random variables, the main result follows by combining the off-diagonal $L_p$ norm bound in Theorem 4 and the diagonal $L_p$ norm bound in Theorem 6 and (51)-(53). ∎

# D  The Azuma-Hoeffding Inequality

A sequence of random variables $Z_1, Z_2, \ldots$, is called a *martingale difference sequence* (MDS) with respect to another sequence of random variables $X_1, X_2, \ldots$, if for any $t$, $E[|Z_t|] < \infty$ and $E[Z_t|X_1, \ldots, X_{t-1}] = 0$ almost surely. By construction, if $X_t$ is a martingale, then $Z_t = X_t - X_{t-1}$ will be a MDS, which explains the name.

Let $\{X_t, t = 0, 1, 2, \ldots\}$ be a martingale sequence, and let $Z_t = X_t - X_{t-1}$ be a MDS. Assume that $Z_t$ is bounded, i.e.,

$$
|Z_t| = |X_t - X_{t-1}| < c_t \,, \tag{79}
$$

and let $\mathbf{c} = [c_1 \ \ldots \ c_T]$ be the vector of the upper bounds. Then, the Azuma-Hoeffding inequality states that: for any $\tau > 0$,

$$
P\left( \left| \sum_{t=1}^T Z_t \right| \geq \tau \right) \leq 2 \exp\left\{ -\frac{\tau^2}{2\|\mathbf{c}\|_2^2} \right\} \,. \tag{80}
$$

For the special case when $c_t = c$, the bound can be simplified to: for any $\epsilon > 0$

$$
P\left( \frac{1}{T} \left| \sum_{t=1}^T Z_t \right| \geq \epsilon \right) \leq 2 \exp\left\{ -T \frac{\epsilon^2}{2c^2} \right\} \,. \tag{81}
$$

The result can be extended to the setting of general subGaussian tails for $Z_t$, e.g., see [39], and also applies to general MDSs $Z_t|\mathcal{F}_{t-1}$ where $\{\mathcal{F}_t\}$ is the filtration.

# E  The Azuma-Bernstein Inequality

Let $\{X_t, t = 0, 1, 2, \ldots\}$ be a martingale sequence, and let $Z_t = X_t - X_{t-1}$ be a MDS. However, we now assume that $Z_t|X_1, \ldots, X_{t-1}$ has a sub-exponential tail, so that

$$
P(|Z_t|X_1, \ldots, X_{t-1}| \geq \tau) \leq 2 \exp(-c\tau/\kappa) \,, \tag{82}
$$

where $\kappa = \||Z_t|X_1, \ldots, X_{t-1}\|_{\psi_1}$ is the sub-exponential norm or $\psi_1$ norm [44]. Then, we have the following result:

**Theorem 8** *Let $\{Z_t\}$ be a MDS which satisfies (82). Then, for every $\mathbf{a} = [a_1 \; \ldots \; a_T] \in \mathbb{R}^T$, for any $\tau > 0$, we have*

$$P\left(\left|\sum_{t=1}^{T} a_t Z_t\right| \geq \tau\right) \leq 2\exp\left\{-\min\left(\frac{\tau^2}{4c\kappa^2\|\mathbf{a}\|_2^2}, \frac{\eta\tau}{2\kappa\|\mathbf{a}\|_\infty}\right)\right\}, \tag{83}$$

*for absolute constants $c, \eta > 0$. In particular, for any $\epsilon > 0$, for a constant $\gamma > 0$, we have*

$$P\left(\frac{1}{T}\left|\sum_{t=1}^{T} Z_t\right| \geq \epsilon\right) \leq 2\exp\left\{-\gamma T \min\left(\frac{\epsilon^2}{\kappa^2}, \frac{\epsilon}{\kappa}\right)\right\}. \tag{84}$$

*Proof:* Recall that if $Y$ is a sub-exponential random variable, then the moment-generating function (MGF) of $Y$ satisfies the following result [44, Lemma 5.15]: For $s$ such that $|s| \leq \eta/\kappa_1$, we have

$$E[\exp(sY)] \leq \exp(cs^2\kappa^2), \tag{85}$$

where $\kappa_1 = \|Y\|_{\psi_1}$ and $\eta, c$ are absolute constants. In particular, since $Z_t|\mathcal{F}_{t-1}$ are subexponential, with $\kappa_1 = \max_t \||Z_t|X_1,\ldots,X_{t-1}\||_{\psi_1}$, for $|s| \leq \eta/\kappa_1$, we have

$$E_{X_t|X_1,\ldots,X_{t-1}}[\exp(sZ_t)] \leq \exp(cs^2\kappa_1^2), \quad \forall t. \tag{86}$$

For any $s > 0$, note that

$$P\left(\sum_{t=1}^{T} a_t Z_t \geq \tau\right) = P\left(\exp\left(s\sum_{t=1}^{T} a_t Z_t\right) \geq \exp(s\tau)\right) \leq \exp(-s\tau)E\left[\exp\left(s\sum_{t=1}^{T} a_t Z_t\right)\right]. \tag{87}$$

For $|s| \leq \eta/(\kappa_1\|\mathbf{a}\|_\infty)$ so that $|a_t s| \leq \eta/\kappa_1$ for all $t$, the expectation can be bounded using (86) as follows:

$$E\left[\exp\left(s\sum_{t=1}^{T} a_t Z_t\right)\right] = E_{(X_1,\ldots,X_T)}\left[\prod_{t=1}^{T} \exp(sa_t Z_t)\right]$$

$$= E_{(X_1,\ldots,X_{T-1})}\left[E_{X_T|X_1,\ldots,X_{T-1}}[\exp(sa_T Z_T)]\prod_{t=1}^{T-1} \exp(sa_t Z_t)\right]$$

$$\leq \exp(cs^2 a_T^2 \kappa^2)E_{(X_1,\ldots,X_{T-1})}\left[\prod_{t=1}^{T-1} \exp(sa_t Z_t)\right]$$

$$\leq \exp(cs^2 a_T^2 \kappa^2)\exp(cs^2 a_{T-1}^2 \kappa^2)E_{(X_1,\ldots,X_{T-2})}\left[\prod_{t=1}^{T-2} \exp(sa_t Z_t)\right]$$

$$\cdots$$

$$\leq \exp\left(cs^2\kappa^2\|\mathbf{a}\|_2^2\right).$$

Plugging this back to (87), for $|s| \leq \eta/\kappa$, we have

$$P\left(\sum_{t=1}^{T} a_t Z_t \geq \tau\right) \leq \exp(-s\tau + cs^2\kappa^2\|\mathbf{a}\|_2^2). \tag{88}$$

Choosing $s = \min\left(\frac{\tau}{2c\kappa^2\|\mathbf{a}\|_2^2}, \frac{\eta}{\kappa\|\mathbf{a}\|_\infty}\right)$, we obtain

$$P\left(\sum_{t=1}^{T} a_t Z_t \geq \tau\right) \leq \exp\left\{-\min\left(\frac{\tau^2}{4c\kappa^2\|\mathbf{a}\|_2^2}, \frac{\eta\tau}{2\kappa\|\mathbf{a}\|_\infty}\right)\right\}. \tag{89}$$

Repeating the same argument with $-Z_t$ instead of $X_t$, we obtain the same bound for $P(-\sum_t a_t Z_t \geq \tau)$. Combining the two results gives us (83).

Now, with $a_t = 1, t = 1, \ldots, T$, and $\tau = T\epsilon$ in (83), for a suitable constant $\gamma > 0$ we have

$$P\left(\frac{1}{T}\left|\sum_{t=1}^{T} Z_t\right| \geq \epsilon\right) \leq \exp\left\{-\gamma T \min\left(\frac{\epsilon^2}{\kappa^2}, \frac{\epsilon}{\kappa}\right)\right\}. \tag{90}$$

712 That completes the proof. ∎

713 The result can also be stated in terms of a general sub-exponential MDS $Z_t|\mathcal{F}_{t-1}$, where $\{\mathcal{F}_t\}$ is the
714 filtration.