[Reviews · NeurIPS 2019]

Reviewer 1



Originality: I believe the results in the paper are novel, I am not aware of other works for uniform large deviation bounds for random quadratic forms, where the random entries form a MDS. The main tools used - namely decoupling and chaining - are standard tools for deriving such bounds when the random entries are independent, but the paper extends them in a novel way to handle MDSs. Clarity: I found the paper to be well written in general, with the notation well defined, and was able to follow the arguments in the main text. I have some minor comments regarding typos and some technical questions. The list of typos mentioned below is not complete and I would suggest that the authors make another careful pass of the paper (including the supplementary material) to flush out other typos. 1) In line 30, it would be good to explain what the notation “vec(A)” means (i.e., how is the vector constructed exactly). 2) Line 82: typo in definition of (2,\infty) norm. 3) Line 84: “second class in..” -> “second class is..” 4) Line 85: “literature have used…” -> “literature has used…” 5) In lines 94-96, you explain that Hanson Wright inequality is for a fixed matrix, while you consider a uniform bound over a set of matrices. Still, if \mathcal{A} is a compact set, couldn’t one use a naive union bound argument over a cover of \mathcal{A}? 6) Line 99: empty bracket [] appears after [19,31]. 7) Line 147: missing comma between references. 8) Line 156: I could not verify why \eta_j is a sub-Gaussian MDS (I can see the sub-Gaussian part, but not the MDS-2 condition), this could be explained in a footnote. Same comment for \eta_j in line 164. 9) Line 158: “…be an bounded” -> “…be a bounded…” 10) Line 199: I don’t understand why there is a sup and inf (over u \in \mathcal{A}) in the definition of RIP? Moreover, \delta_s is defined as the smallest constant in (0,1) for which eq. (22) holds (hence it is well defined), see for eg. the book “Mathematical Introduction to Compressive Sensing” by Foucart and Rauhut for a precise definition. 11) Corollary 1 has the notation vec(X) in the statement which has not yet been defined. 12) In line 227, shouldn’t we have d_{F}(\mathcal{A}) = 1? Similarly, shouldn’t we have d_{2->2} (\mathcal{A}) = 1/\sqrt{n} ? Also, in line 230, \norm{\theta}_2^2 should be replaced with 1. It would also help to inform the reader that the unit norm assumption is w.l.o.g. 13) Line 235: should be “…sequence of x_i’s…”; line 238 should be: “… our result provides a tool…” ; line 239: should be “…that is obtained…” 14) Line 242: should be “…and are a desirable alternative to random matrices with i.i.d entries…” 15) Line 253 has an incomplete sentence; this needs to be fixed. Quality: I haven’t been able to check the proofs in detail as they are all confined to the supplementary material and are fairly long and tedious. It will certainly help if the (top level) proof outline for bounding the L_p norms of the two terms in eq. (6) could be provided in the main text. I feel this is important, as this is really the main crux of this paper. Significance: I feel that the bounds obtained in this paper have the potential of being useful in a number of problem settings where the data is obtained sequentially or adaptively, and thus contains statistical dependencies with the past observations. So, I am quite positive that the results will be of interest to researchers in high dimensional statistics, compressed sensing and randomized linear algebra.

Reviewer 2



This article studies concentration (or large deviation) inequalities for quadratic forms involving vectors with a martingale difference dependence structure. The article is overall well written, the results most likely exact. Yet, I have several important concerns: * on relevance: the arguments offerred by the authors on practical use of their results are vague and the concrete examples (the corollaries) are clearly unconvincing, if not artificial. The authors really need to dig this question as there seems to be no obvious case of evident interest in the standard literature. * technically speaking, there appears to be little fundamentally new in the derivations of the main results. Worse, in my opinion, the authors concentrate their efforts in quite obvious preliminaries (around page 3 and 4) only to scheme quickly over the actual key innovative arguments (deferred to appendices). There also appears to be essentially no difference between the iid and MDS setting, so what's the conclusion? * in terms of adequation to NIPS, I find the article more of a (not substantial) mathematical exercise providing an epsilon-improvement over existing results with no prior thoughts on the entailing machine learning consequences. The keywrod "adaptive learning" raised by the authors in introduction and conclusion is possibly a good anchor, but this is as far as the reflexion goes unfortunately. For these reasons, I believe the work cannot be math-wise nor ML-wise considered at the demanding level of NIPS. I: - "The RIP is a well known ... result of this type, which ... major impact" and then "The JL is ... well known result of this type, which has major..." --> please be more inventive in phrasing. - the word "such" is (unbearably) used in almost every sentence. Please avoid these repetitions. - "to get the large deviation bound" --> which large deviation bound? This has not been introduced yet. - "existing analysis for specific cases goes to great lengths to work with or around such dependence" --> I do not understand this sentence, please restate. - "For decoupling, we present a new result on decoupling" --> restate II: - reference missing after [19],[31] - the discussion on converting zeta into a random matrix X, etc. is redundant with the previous section. - MDS acronym already introduced - the sentence preceding (4)-(5) ("we decompose the quadratic form") could implicitly suggest that C_A=B_A+D_A which obviously is not the case. A more appropriate statement would avoid this possible confusion. Also, a quadratic form per se has no "diagonal" or "off-diagonal" term; so this makes little sense anyways. - "the off-diagonal terms of E|Ax|² is 0" makes little sense (this is a scalar, not a matrix) - the result (6) is somewhat straightforward and could have been schemed over more quickly. On the opposite (7) is more interesting and would have deserved a better, more extensive treatment (rather than delegating it to Appendix). In particular, it would be appropriate to explain here why each has a different order of magnitude in 'p'. - I suppose 'a', 'b' and 'c' are independent of 'p'? - why |C_A|? C_A is positive by definition. - "A naive approach..." --> which approach? - the way the section is organized is difficult to follow. If I understand correctly: first we decompose C_A into B_A and D_A, to show that the final result is (9)... proved in Appendix B and C. Then, why get into gamma-functions here? Are we detailing a specific argument to reach (9)? Please help the reader through in the way the results are introduced, and go sequentially from arguments to conclusions, not the other way around. - "an bounded" - "such an approach needs to first generalize several keys results..." --> why not do it? - "We have also... which is also... It is also" --> please check these repetitions III: - I do not understand the practical relevance of Corollary 1. In which context would the matrix X be organized with tis "vec-wise" MDS property? The provided example is not convincing. Which concrete scenario do you have in mind? Please defend your case here. - "with iid entry" --> entries - the introduction of the Toeplitz matrices is a bit naiv. Toeplitz matrices are not a "desirable alternative" to matrix with iid entries. There's little in common between these objects... - again, please justify the practical relevance of Corollary 2. In which context zeta could form an MDS? To me, Toeplitz matrices are mostly used to model the coefficients of stationary processes which, by definition, are immuable with time and do not depend on each other, certainly not at least with an MDS structure. So what again do you have in mind here? - am I wrong to say that (28) is simply saying that |Xu|²=|u|² for all u in A? - "for the iid case" --> full stop missing - I do not see why vec(X) satisfies an MDS in CountSketch. The constraint on the eta_ij's is not "sequential". As you state, each row entry depends on all other entries (for indices below and above), so it's not an MDS to me. Could you clarify this point? IV: - "over the past few decades" --> I wouldn't go that far. - "We anticipate our results to simplify and help make advances in analyzing learning settings based on adaptive data collection" --> ok, but how precisely? This argument from the introduction is replicated here with no further argument. I do not see in which context precisely and the corollaries provided in the core of the article are not convincing. Please elaborate. - "sharpen our analysis" --> what do you mean? In which direction? Do you intend to get rid of the log(n) terms? If so, I do not see this as a worthwhile ambition. - all in all, I was expecting to see in this conclusion a real opening to the practical relevance of the article findings, but I'm still waiting for it.

Reviewer 3



In this paper, the authors prove a concentration result for quadratic forms, which removes the usual iid assumption from the random variables, and replace it with an "MDS" assumption (each new variable is sub-gaussian and has zero mean when conditioned to the previous ones). Their proof uses generic chaining. Then, they apply this result to proving a new version of the Restricted Isometry Property, and the Johnson-Lindestrauss lemma. They also provide some special instantiations in each case. I liked this paper. The writing is pretty clear, if sometimes a bit verbose (some things are repeated quite a few times, eg line 116-118), and the exploration of the topic is thorough. Despite the many works around concentration of measure and chaining, as far as I can tell the proposed bound is new, however it is possible that some work that I am not aware of may be similar in the literature (which is, again, very large). The new bounds for the RIP and JL may be re-used by practitioners in data compression and machine learning, since they are at the core of many analysis. The authors especially emphasize the use in a sequential context. A possible critic would be that it is not very clear for the reader how "natural" or "applicable" the MDS assumption is, and it can sometimes give the impression that the authors slightly oversell how much of a "substantial generalization" departing from a "key limitation" it is in practice (even if it might be mathematically speaking). The authors briefly mention bandits or active learning, would it be possible to describe a bit more these contexts, in a few equations, and emphasize how an MDS system would appear ? This may be more useful and convincing for the reader (especially from these communities) than some examples (eg RIP Toeplitz), if space is needed. Minor comments / typos: - line 3 : the the - line 99 : missing ref - maybe (6) should be encapsulated in a small Lemma, and what precedes (simple computations) in the proof ? - both historically and mathematically, it can be argued that the JL lemma precedes the RIP (in fact, the RIP can be seen as a "uniform" version of JL for sparse vectors, and the classical proof of the RIP is based on JL and a covering argument). So it would make sense to invert the two sections on RIP and JL - the results for the RIP are corollaries, and the one for JL is a lemma, is there a reason for this ? It can be a bit confusing. - Even though the countsketch is based on JL, it is always mentioned as a separate example, it can also be a bit confusing **EDIT after feedback** I thank the authors for their feedback, which I find convincing on the points that I raised. I stand by my score, however I encourage the authors to take into account the critics raised in the other reviews, some of which I find legitimate.

[Author Response · NeurIPS 2019]

We appreciate the detailed, insightful, and encouraging comments from the revewiers, as well as the constructive
criticism. We first highlight the novelty of the results and analyses and discuss an use case for adaptive learning where
the results will be drectly applicable. Subsequently, we respond to specific comments from individual reviewers.

**New results and analyses.** As the reveiwers noted, the main technical results (Theorems 1 and 2) are new. The fact that
the results match the corresponding results for the i.i.d. case is desirable, as this will allow extension of existing results
from the i.i.d. to the MDS case in a seamless manner. However, the technical details rely on several new results for
MDSs. First, for decoupling, the i.i.d. case is somewhat straightforward since one has to handle products of independent
random variables (r.v.s). For MDSs (Appendix A), we worked with a quadratic form of dependent r.v.s, had to first
show distributional equivalence of two such forms, and finally got the decoupling result by using a *decoupled tangent*
*sequence* to the original MDS, but *not an independent MDS* (Theorem 3). Second, for uniformly bounding the $L_p$
norms of r.v.s from the MDS quadratic form, we showed that or both sub-Gaussian and sub-exponential tails $E[\sup \cdots]$
can be upper bounded by $\sup E[\cdot]$, which is easier to handle, and an additive term which depends on Talagrand's
$\gamma_2$ function. The analyses (Lemma 7 and 8) utilize the core argument in generic chaining along with concentration
bounds (Azuma-Hoeffding and Azuma-Bernstein) for MDSs, leading to new results on uniform bounds on $L_p$ norms of
quadratic forms of MDSs (Theorems 4 and 5). We relegated most of such technical details to the appendix, but based
on the constructive feedback we will discuss the core ideas behind these results and proofs in the main paper.

**Use case for adaptive learning: Linear contextual bandits.**    Our results on Restricted Isometry Property (RIP)
have direct applications to many adaptive learning problems, including linear contextual bandit (LCB) learning. As
a concrete example, consider the smoothed analysis model for LCB proposed in [1, 2]. The parameter estimation
step involves solving an ordinary least squares (OLS) problem with a design matrix whose rows are sub-Gaussian
MDS. The parameter estimation error rate depends on the minimum eigenvalue of the design matrix. Our results
significantly simplifies the minimum eigenvalue analysis (Lemma B.1 of [1]), which currently uses cumbersome
high-probability boundedness arguments to satisfy the boundedness requirement in [3]. In fact, our results provide a
tighter high-probability bound on minimum eigenvalue of the design matrix. Moreover, our RIP results can be directly
applied to the high-dimensional LCB setting where the latent parameter is assumed to have structure (such as sparsity)
and parameter recovery requires RIP of the design matrix. We will expand on this LCB application in Sec. 3.

**R1.** We will include a brief sketch of how the terms in (6) are bounded and give some details on what $a, b, c$ are in (7).
We appreciate the detailed comments and will update the draft to address these. Brief responses for some of the points
raised: for compact sets, one can indeed use a covering argument along with Hanson-Wright, but generic chaining
gives a sharper bound by using a hierarchical covering argument; lines 156, 164, its a typo, the expectation should be a
conditional expectation, the analysis in Appendix B uses the correct form, we will fix it; line 199, we will update it to
be $\forall u \in A$, the inf-sup form is sometimes used in high-d statistics; line 227, you are correct, we will fix them.

**R3.** First, we give a concrete example above on LCB [1] where the main results can be directly used. Second, while the
results for the i.i.d. and MDS cases match, note that the MDS results needed new tools and results which we developed
as part of the work. For example, consider the decoupling results in Appendix A. Recall that for decoupling for the
i.i.d. setting, one just needs to consider an independent copy of the random vector. However, for the MDS setting, an
independent copy of the MDS *does not* lead to decoupling, so we had to develop the MDS decoupling result based on a
decoupled tangent sequence. Similar new results were developed in the context of generic chaining. We appreciate
the detailed comments, we will make a pass on the paper to incorporate these. Brief responses for some of the points
raised: for (7), we plan to bring the technical results (Theorems 4 and 5 in Appendix B) in the main paper; $a, b, c$ are
independent of $p$; yes, we needed to introduce the $\gamma_\beta$ function because $a, b$ in (9) depend on the $\gamma_2$ function, this can be
seen by comparing (9) with Theorems 1 and 2, but we will work on the writeup to make these clear; we in fact now
have the sharper analysis which drops the extra $\log n$ term; for Corollary 1, as we discuss above, the vec-wise MDS
shows up for LCB [1]; existing results on RIP for Toeplitz matrices rely on the $(2p - 1)$ elements being drawn i.i.d.,
and we extend the result to MDSs, but we plan to replace the Toeplitz example with the arguably more compelling LCB
example; for CountSketch, you are right, each $\eta_{ij}$ is not sampled sequentially, but vec$(X)$ is still a MDS since the
Rademacher r.v.s $\delta_{ij}$ are independent of $\eta_{ij}$, the conditional expectation $E[\delta_{ij}\eta_{ij}|.] = E[\delta_{ij}]E[\eta_{ij}|.] = 0$.

**R4.** We have discussed a concrete example on linear contextual bandits [1] where the main results can be directly used.
Brief responses for some of the points raised: we will state (6) as a Lemma, and briefly sketch how the terms in (6) are
bounded (in Appendix B); we agree with the historical remark, we will switch the RIP and JL sub-sections, make the
presentation uniform (all Corollaries), and also add additional remarks on the countsketch example. We appreciate the
detailed comments and will update the draft based on these.

[1] Kannan et al., A Smoothed Analysis of the Greedy Algorithm for the Linear Contextual Bandit Problem, 2018.

[2] Raghavan et al., The Externalities of Exploration and How Data Diversity Helps Exploitation, 2018.

[3] Tropp, J. A. User-friendly Tail Bounds for Sums of Random Matrices, 2012.


[Meta-Review · NeurIPS 2019]

Random quadratic forms have been a subject of interest in the community, and the paper makes notable technical contributions appreciated by the reviewers. On the flip side MDS is a very specific form of dependence, and the presentation focuses almost exclusively on probabilistic aspects, so that the paper can appear niche to the NeurIPS public. Overall the paper is recommended for acceptance in NeurIPS, but it is strongly advised that the authors develop the application (convincingly mentioned in the rebuttal) to linear context bandits and possibly emphasize this motivation up front to increase visibility and interest in the context of NeurIPS.